# Functional and structural characterization of a flavoprotein monooxygenase essential for biogenesis of tryptophylquinone cofactor

Toshinori Oozeki[1,3], Tadashi Nakai [1,2,3], Kazuki Kozakai[1], Kazuki Okamoto[1], Shun'ichi Kuroda[1], Kazuo Kobayashi[1], Katsuyuki Tanizawa[1] & Toshihide Okajima [1✉]

Bioconversion of peptidyl amino acids into enzyme cofactors is an important post-translational modification. Here, we report a flavoprotein, essential for biosynthesis of a protein-derived quinone cofactor, cysteine tryptophylquinone, contained in a widely distributed bacterial enzyme, quinohemoprotein amine dehydrogenase. The purified flavoprotein catalyzes the single-turnover dihydroxylation of the tryptophylquinone-precursor, tryptophan, in the protein substrate containing triple intra-peptidyl crosslinks that are pre-formed by a radical S-adenosylmethionine enzyme within the ternary complex of these proteins. Crystal structure of the peptidyl tryptophan dihydroxylase reveals a large pocket that may dock the protein substrate with the bound flavin adenine dinucleotide situated close to the precursor tryptophan. Based on the enzyme-protein substrate docking model, we propose a chemical reaction mechanism of peptidyl tryptophan dihydroxylation catalyzed by the flavoprotein monooxygenase. The diversity of the tryptophylquinone-generating systems suggests convergent evolution of the peptidyl tryptophan-derived cofactors in different proteins.

---

[1] Institute of Scientific and Industrial Research, Osaka University, Osaka, Japan. [2] Faculty of Life Sciences, Hiroshima Institute of Technology, Hiroshima, Japan. [3] These authors contributed equally: Toshinori Oozeki, Tadashi Nakai. ✉email: tokajima@sanken.osaka-u.ac.jp

Posttranslational protein modifications expand the chemical repertoire of amino acid residues defined by genetic codons. A variety of protein-derived cofactors have so far been identified at the active sites of many enzymes, and their biosynthesis from one or more amino acid residues is recognized as an important category of posttranslational modification[1–3]. Among them, the quinone cofactors produced from aromatic amino acid residues, Tyr or Trp, and, in some cases, attached covalently to another residue, play redox catalytic roles[4–6]. These cofactors are produced either by an autocatalytic process, assisted only by the proper protein fold and occasionally a metal ion, or by the participation of one or more modifying enzymes[1–3].

Cysteine tryptophylquinone (CTQ) is a protein-derived quinone cofactor initially identified in quinohemoprotein amine dehydrogenase (QHNDH), a bacterial enzyme that catalyzes oxidative deamination of various aliphatic primary amines for use as energy, carbon, and nitrogen sources[7,8]. The crystal structures of QHNDH, determined for the enzymes from two different Gram-negative bacteria[9,10], revealed the common heterotrimeric subunit structure (Fig. 1a), consisting of the ~60-kDa α-subunit that contains two *c*-type hemes, the ~37-kDa β-subunit, and the ~9-kDa γ-subunit that contains CTQ in an uncommon protein structure with four intra-peptidyl thioether bonds (three Cys-to-Asp/Glu crosslinks, and one in CTQ) (Fig. 1b). The intricate structure of the γ-subunit as well as the presence of CTQ indicates that multiple steps of posttranslational modification are required for the generation of CTQ in the mature γ-subunit.

Structural genes encoding QHNDH constitute an operon, termed *qhp*, along with several nearby genes (*qhpABCDEFGR*; the arrangement, order, and coding strands of the genes are variable), all of which are necessary for the amine-induced expression of the enzyme in the periplasm of bacterial cells[11]. The *qhp* operon is distributed in >1300 bacterial species, currently identified by the position-specific iterated BLAST search (https://blast.ncbi.nlm.nih.gov/Blast.cgi)[12]. The *qhpA*, *qhpB*, and *qhpC* genes encode the α-, β-, and γ-subunits of QHNDH, respectively. The *qhpD* gene encodes an unusual radical *S*-adenosylmethionine (SAM) enzyme (QhpD) that catalyzes sequential formation of three Cys-to-Asp/Glu thioether bonds within a single polypeptide of QhpC

(γ-subunit)[13,14]. The *qhpE* gene encodes a subtilisin-like serine protease (QhpE) that cleaves the N-terminal 28-residue leader peptide from the crosslinked QhpC[15] (Fig. 1b) before periplasmic translocation through an efflux ABC transporter, encoded by the *qhpF* gene[11]. The QhpE protease is also unusual in that it serves as a single-turnover processing enzyme acting in a suicidal manner. Here, we shed light on the *qhpG* gene, which has been predicted to encode a flavoprotein monooxygenase, directly involved in CTQ biogenesis[11]. Biochemical evidence reported herein proves that QhpG is an atypical single-component monooxygenase, catalyzing dihydroxylation of an unmodified Trp residue in the protein substrate.

## Results

**Analysis of quinone-less γ-subunit in inactive QHNDH.** We previously demonstrated that inactive QHNDH, produced in the *qhpG* gene-disrupted mutant strain (Δ*qhpG*) of *Paracoccus denitrificans*, contains no quinone group in the γ-subunit (fully processed QhpC)[11]. To elucidate the modification state of the CTQ-precursor Trp residue in the quinone-less QhpC, we isolated the QhpC polypeptide in the inactive QHNDH complex from the periplasm of the Δ*qhpG* mutant by Ni affinity chromatography utilizing N-terminally hexa-His (His₆)-tagged α-subunit (QhpA), and analyzed it by matrix-assisted laser desorption ionization-time of flight (MALDI-TOF) mass spectrometry (MS). The observed mass (*m/z*, 8829.3) was smaller by ~28 mass unit than the calculated mass of the γ-subunit (*m/z*, 8857.6)[13], and corresponded well to that of the QhpC polypeptide (*m/z*, 8828.6) containing three intra-peptidyl thioether bonds formed between Cys and Asp or Glu residues, and each one of the unmodified Cys and Trp residues (without two oxygen atoms and a Cys–Trp crosslink contained in CTQ of γ-subunit) (Supplementary Fig. 1a). In addition, treatment of the QhpC sample with 2-iodoacetamide (IAA) shifted the mass spectrum peak to a higher *m/z* value (*m/z*, 8887.7), the increase (Δ = ~58) corresponding to acetamidation of a single free Cys residue. Although the MS analysis of the whole peptide is insufficient for identification of the unmodified residues, Cys37 and Trp43 that form CTQ most likely

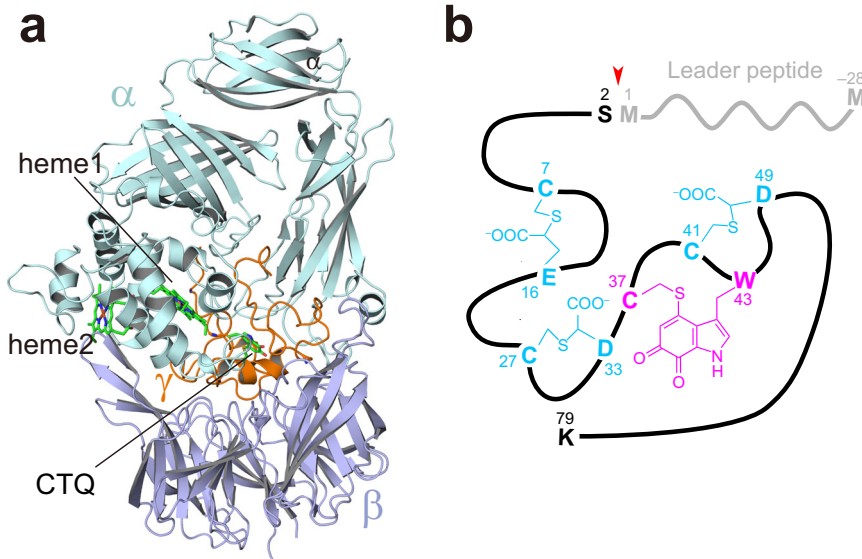

**Fig. 1 Crystal structure of QHNDH and schematic representation of γ-subunit. a** Overall structure of QHNDH from *Ps. putida*. The α-subunit (QhpA, pale cyan), β-subunit (QhpB, purple), and γ-subunit (QhpC, orange) are depicted by a cartoon model with two hemes and CTQ shown in green stick model. **b** Schematic presentation of γ-subunit polypeptide with a 28-residue leader peptide (light gray). Chemical structures of thioether crosslinks and CTQ are shown in cyan and magenta, respectively. The QhpE-cleavage site is indicated by a red arrowhead.

remain intact in the quinone-less QhpC (Supplementary Fig. 1b). Thus, it is strongly suggested that QhpG is an enzyme that acts on the quinone-less QhpC with three intra-peptidyl thioether bonds as the protein substrate, and converts it into a mature form (γ-subunit) containing CTQ, or its immediate precursor, such as mono- or di-hydroxylated Trp.

**Purification and characterization of QhpG**. Our initial attempt at purifying QhpG from *Pa. denitrificans* was unsuccessful due to the formation of inclusion bodies during expression in recombinant *Escherichia coli* cells. Therefore, we subsequently cloned an ortholog of QhpG from *Pseudomonas putida*, which also produces QHNDH[7,10]. The *Ps. putida* QhpG expressed in recombinant *E. coli* cells was purified to homogeneity in a soluble form (Supplementary Fig. 2a). It behaved as a monomeric protein of molecular weight (MW) of ~40,000 in a gel-filtration chromatographic analysis (calculated MW, 47,194), and exhibited a ultraviolet–visible spectrum characteristic of an FAD-containing protein, which was readily reduced with sodium dithionite under anaerobic conditions (Supplementary Fig. 2b). The addition of $O_2$-saturated buffer to the dithionite-reduced QhpG resulted in rapid re-oxidation of the bound FAD (Supplementary Fig. 3a). FAD was found to be tightly, but non-covalently bound to QhpG in a molar ratio of nearly 1:1 (determined spectrophotometrically), as it was extractable by heat treatment at 50 °C for 10 min (Supplementary Fig. 2c). For studying the QhpG reaction in vitro, we also cloned *QhpC* and *QhpD* genes of *Ps. putida*, and expressed it as a stable QhpCD binary complex in *E. coli* cells, as described previously for *Pa. denitrificans* proteins[14]. In the following experiments, the QhpCD complex derived from *Ps. putida*, in which QhpC is the nascent polypeptide carrying the 28-residue leader peptide that is necessary for the interaction with QhpD and a C-terminal Twin-Strep ($St_2$)-tag (hereafter designated linear QhpC; calculated MW, 14,875.5), was used after chemical reconstitution of [4Fe–4S] clusters contained in QhpD[14].

**Preparation of substrate for QhpG**. During the preparation of the protein substrate for QhpG, that is, the quinone-less QhpC containing three intra-peptidyl thioether bonds, 28-residue leader peptide, and one each of free Cys and Trp residues (Cys37 and Trp43) (hereafter designated crosslinked QhpC; calculated MW, 14,869.4), we found that the QhpD-catalyzed thioether bond formation in QhpC was significantly promoted by QhpG. Thus, in the presence of an equimolar amount of QhpG, the crosslinked QhpC was formed almost completely (Fig. 2a, top panel), whereas, in the absence of QhpG, the linear QhpC underwent only partial formation of thioether bonds with 1–4 Cys residues, which remained modifiable with IAA (Fig. 2a, bottom). These results suggest that QhpG interacts with the QhpCD binary complex. Indeed, as shown in Fig. 2b, mobility shift assays by native polyacrylamide gel electrophoresis (PAGE) performed under an anaerobic condition revealed the formation of a QhpCDG ternary complex (middle band) between the bands of the QhpCD binary complex (lower) and QhpG (upper), depending on the increasing amounts of QhpCD in combination with a constant amount of QhpG. Moreover, the formation of the ternary complex was more prominent with the crosslinked QhpC (lanes 7–10) than with the linear QhpC (lanes 2–5). These results show that QhpG binds the QhpCD complex more preferentially through the crosslinked QhpC than the linear one lacking internal thioether bonds. Presumably, the QhpD-catalyzed thioether bond formation in QhpC is facilitated by QhpG that captures the partially crosslinked QhpC, which is structurally more stable than the linear one.

**Analysis of QhpG–QhpC interaction**. The interaction of QhpG with QhpC was further analyzed quantitatively by bio-layer interferometry (BLI) assays, using QhpC polypeptide immobilized on the biosensor surface (Fig. 2c). Linear and crosslinked QhpC polypeptides were isolated from the QhpCD binary complex by removing the QhpD protein by heat denaturation, before and after conducting the crosslinking reaction, respectively. QhpG showed a 100-fold higher affinity for the crosslinked QhpC than the linear one in terms of estimated $K_D$ values (Supplementary Table 1), which agreed with the mobility shift assays (Fig. 2b). Comparison of association ($k_a$) and dissociation ($k_d$) rate constants also indicated faster association and slower dissociation of QhpG for the crosslinked QhpC, explaining the high affinity. Comparable results were obtained in the mobility shift assay on native PAGE, where the interaction of QhpG with the linear QhpC was almost unobservable, in contrast to the significant interaction with the crosslinked QhpC forming a QhpCG binary complex (Fig. 2d). Collectively, it may be concluded that QhpG interacts with the crosslinked QhpC, which serves as the protein substrate.

**Determination of catalytic activity of QhpG**. Assuming that QhpG is an FAD-dependent oxygenase, the purified QhpG was first anaerobically incubated with several reducing reagents: NADPH, NADH, $FADH_2$, and sodium dithionite. Two other small biomolecules, dihydrolipoate and reduced glutathione with lower reduction potentials than FAD, were also tested. Besides the artificial reductant (sodium dithionite) (Supplementary Fig. 2b), none of the physiological reagents reduced the QhpG-bound FAD without affecting its absorption spectrum (Supplementary Fig. 3b–f). Free $FADH_2$ neither reduced nor replaced the bound FAD. Therefore, after reducing the reaction mixture containing the QhpCDG ternary complex with excess sodium dithionite (~3 mM), the single-turnover reaction of QhpG was initiated by the addition of $O_2$-saturated buffer, and continued for 1 h under an atmospheric condition until the initially added dithionite was mostly consumed by air (Supplementary Fig. 4). The reaction product was then precipitated by treatment with cold acetone and digested with Asp-N proteinase, followed by MALDI-TOF MS analysis (Supplementary Fig. 5). Among the peptide fragments produced by Asp-N digestion, the peak labeled d was assigned to the peptide starting at Asp39 and ending at Gln55 in the triply crosslinked QhpD polypeptide, including the CTQ-precursor Trp43 and an internal thioether bond formed between Cys41 and Asp49. Averaged mass ($m/z$, 2054.7 ± 0.2; $n = 10$) of peak d (monoprotonated form) before the QhpG reaction agreed well with the calculated molecular mass ($m/z$, 2054.3) of this peptide (Fig. 3a, top panel). After the QhpG reaction, the relative intensity of peak d decreased significantly, and simultaneously the intensity of a new peak having $m/z$, 2086.7 ± 0.3 ($n = 10$) increased (Fig. 3a, middle panel). The mass increase ($\Delta = 32.0 \pm 0.3$; $n = 10$) was reproducibly observed in QhpG reactions performed with different preparations of the QhpCDG ternary complex, and was consistent with the incorporation of two oxygen atoms into the peak d peptide by the QhpG reaction. Furthermore, when the initial anaerobic reduction with dithionite was done in the $H_2^{18}O$-buffer and then the QhpG reaction was started by the addition of $^{16}O_2$-saturated $H_2^{18}O$ buffer, the reaction product contained two $^{16}O$ atoms but no $^{18}O$ atom (Fig. 3b), supporting that the oxygen atoms inserted into the CTQ-precursor Trp are not derived from solvent $H_2O$. Interestingly, this d + 32 peak was not formed in the QhpG reaction with the crosslinked QhpC alone (Fig. 3a, bottom panel), showing that the QhpG-catalyzed oxygen incorporation proceeds in the QhpCDG ternary complex as efficiently as the QhpD-catalyzed formation of intra-peptidyl

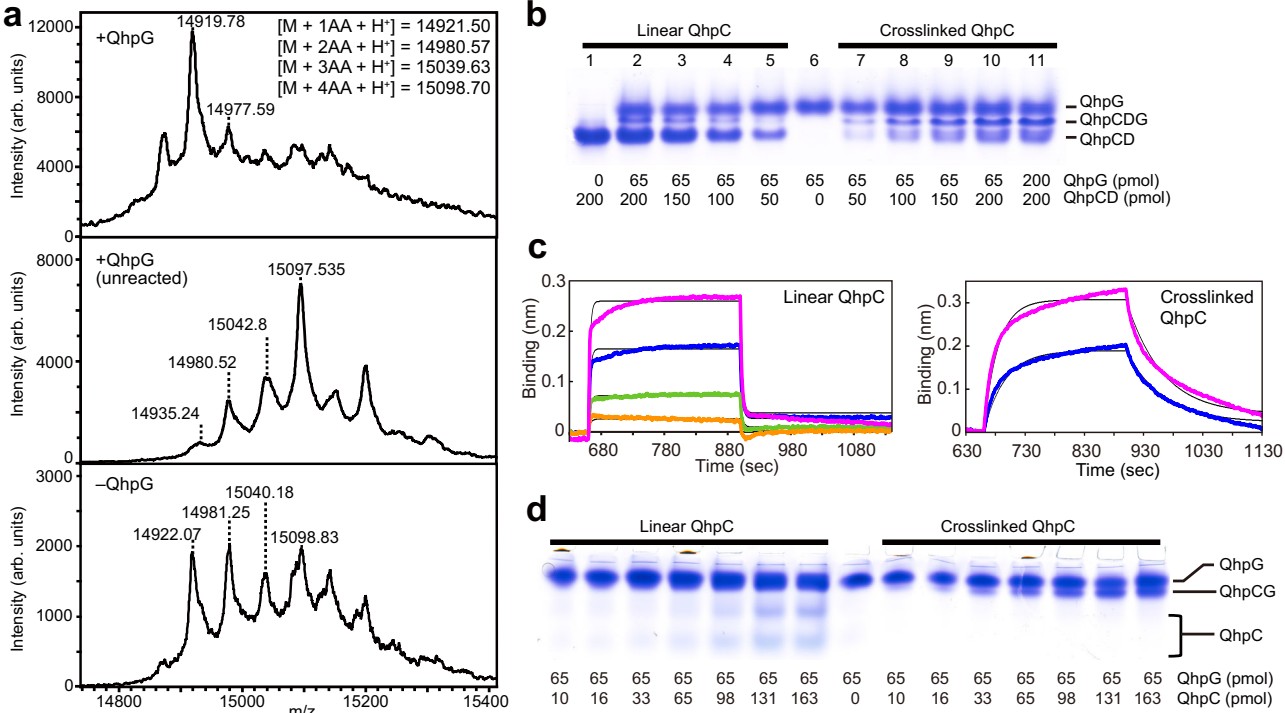

**Fig. 2 Effect of QhpG on QhpD-catalyzed thioether bond formation in QhpC and inter-protein interactions among QhpCDG proteins. a** MALDI-TOF mass spectra are shown for IAA-treated products (crosslinked QhpC) of the QhpD-catalyzed thioether bond formation in the presence of an equimolar amount of QhpG (top panel) and in its absence before (middle) and after (bottom) the QhpD reaction. Inset: Calculated *m/z* values of 1–4 acetamidated (AA) peptides (monoprotonated form). Intensity is expressed in arbitrary units (arb. units) for all mass spectra. Mobility shift assays for interactions between QhpG and the QhpCD binary complex (**b**) and between QhpG and QhpC (**d**). Indicated amounts (pmol) of respective proteins were applied in each lane. In **b**, **d**, the experiments repeated twice independently gave similar results. **c** BLI assays for interactions between QhpG and linear (left) and crosslinked (right) QhpC immobilized on the biosensor tip. The analyte solution (4 µl) contained QhpG at 1.0 µM (magenta), 0.50 µM (blue), 0.25 µM (green), and 0.13 µM (orange) for linear QhpC (left) and at 62.5 nM (magenta) and 31.3 nM (blue) for crosslinked QhpC (right). Binding-induced changes in wavelength (nm) of the transmitted light were recorded for measuring time (s). Thin black curves represent the theoretical fitting of the calculated data (Supplementary Table 1).

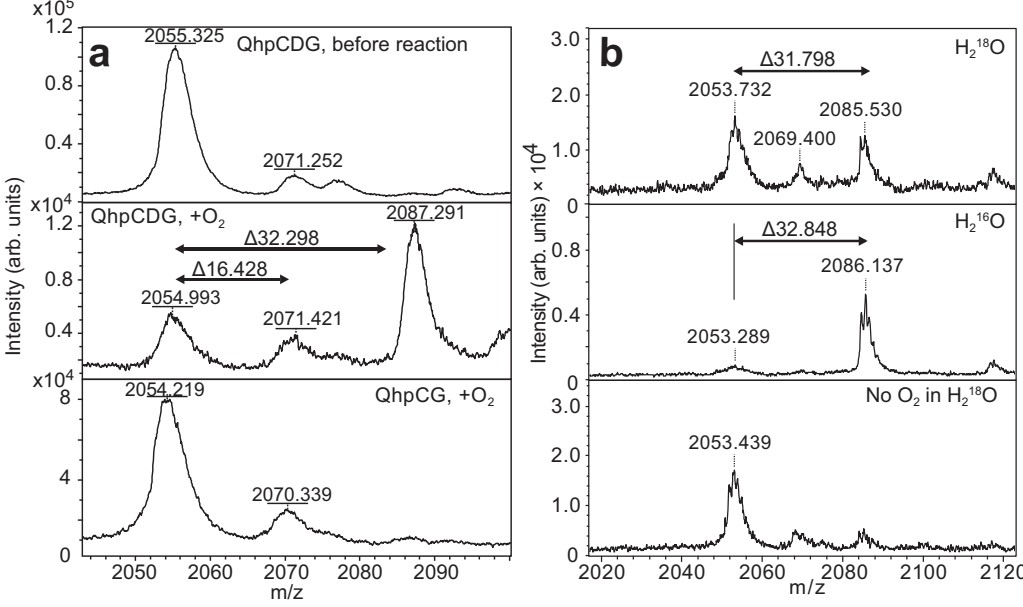

**Fig. 3 Product analysis of QhpG reaction. a** MALDI-TOF mass spectrometric analysis of the QhpG reaction products in the QhpCDG ternary complex before (top panel) and after (middle) addition of O₂-saturated buffer and with the free crosslinked QhpC (bottom) (fragment d of Asp-N digestion). **b** MALDI-TOF mass spectrometric product analysis for the QhpG reactions conducted in H₂¹⁸O-buffer (top panel), H₂¹⁶O-buffer (middle), and without O₂ addition in H₂¹⁸O buffer (bottom).

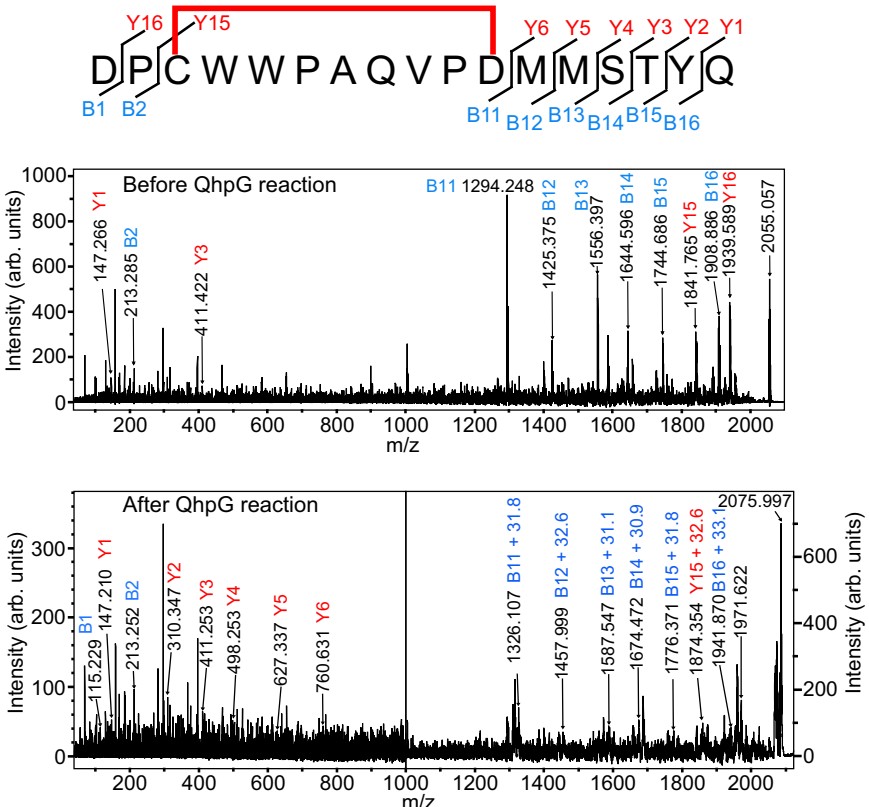

**Fig. 4 MS/MS analysis of peptide fragment produced by QhpG reaction.** Predicted fragmentation pattern of fragment d (Asp39–Gln55) is shown above. MS/MS spectra of the peptides of d and d + 32 peaks obtained before and after the QhpG reaction are shown in upper and lower panels, respectively. Assigned fragments are indicated with *m/z* numbers.

thioether bonds described previously. A minor peak of ~16 mass unit higher than peak d was often observed, suggesting the formation of a reaction intermediate with a single oxygen atom being incorporated into this peptide. However, relative intensities of the d + 16 peak did not change before or after the QhpG reaction; it may have been derived from the partial oxidation of Met residues contained in this peptide, which could occur during Asp-N digestion and sample handling for MS analysis. It is noticeable that the CTQ-forming Cys37 is outside peptide d, indicating that the thioether bond of CTQ was not formed in the peak d peptide. The absence of a dye-stainable quinone group in the product of QhpG reaction (Supplementary Fig. 6) also indicates that tryptophylquinone was not formed. Hence, the 32 mass unit increase may be attributed to the insertion of two oxygen atoms in the form of two hydroxyl groups.

Tandem MS (MS/MS) analysis of the peptides before (peak d) and after the QhpG reaction (peak d + 32) revealed that the incorporation of two oxygen atoms occurred within the circular peptide (Cys41–Asp49), resistant to fragmentation in MS/MS analysis[16] (Fig. 4). Moreover, the d + 32 peak was not present in the reaction with a QhpC mutant, in which Trp43 was replaced by Phe (Fig. 5a). Based on these results, we concluded that the CTQ-precursor Trp43 is doubly hydroxylated, most likely at 6- and 7-positions of the indole ring, by the QhpG reaction. Taken together, QhpG was identified as an unusual flavoprotein that catalyzes dihydroxylation of a peptidyl tryptophan in the single-turnover reaction, although belonging to a monooxygenase as proposed previously[11].

**Crystal structure of QhpG**. We crystallized QhpG as yellowish thin plates (Supplementary Fig. 2d), and determined the crystal

structure at 1.98-Å resolution by single isomorphous replacement with anomalous scattering of a mercury (Hg)-derivatized crystal (Protein Data Bank (PDB) entry ID: 7CTQ, Supplementary Table 2). In the crystal, each asymmetric unit contained two monomers that are related by a non-crystallographic (NCS) two-fold axis and packed compactly (Supplementary Fig. 7a). However, PISA (Proteins, Interfaces, Structures and Assemblies) analysis (https://www.ebi.ac.uk/pdbe/pisa/)[17] indicated that there are no strong interactions enough to form a stable dimer in the protein interface. In addition, MW determination by a gel-filtration method suggested that QhpG is a monomer protein in solution, as described above. Thus, the dimerization is assumed to be a crystallographic artifact. Each monomer consists of a large N-terminal catalytic domain (residues 1–344) and a small C-terminal winged-helix (WH) domain (residues 349–429), with a large pocket between the two domains (Fig. 6a).

The overall structure of QhpG resembled FAD-dependent halogenases[18–20] and monooxygenases[21,22], with the highest structural similarity to *Streptomyces venezuelae* alkylhalidase SvCmlS[20] (PDB entry ID, 3I3L; root-mean-squared deviation, 3.2 Å over 337 superposed residues; sequence identity, 19%; Z-score, 34.1, in Dali server search[23]). The catalytic domain of QhpG contains a glutathione reductase-type Rossmann-fold for FAD binding, in which FAD is bound between two lobes of the catalytic domain (Supplementary Fig. 7b), showing a clear $F_o–F_c$ omit map for the entire FAD molecule (Fig. 6b). Briefly, the isoalloxazine ring of FAD is sandwiched by the side chains of Pro267 and Glu42/Arg240 from the *re-* and *si*-faces, respectively (Fig. 6b). The di-phosphate moiety interacts with the side chains of Arg36 and Arg89, and the main chain NH groups of Ala14 and Asp260. The adenine ring and ribose moieties are packed with Glu34, Arg36, Arg111, Arg140, Gln143, and Val112 with its

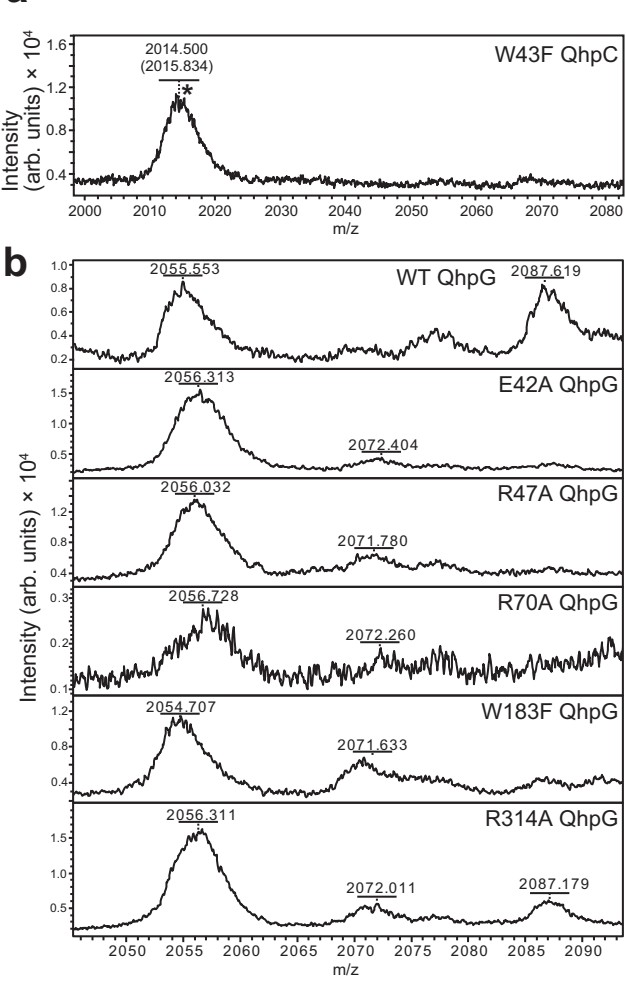

**Fig. 5 MALDI-TOF MS analysis of QhpG-catalyzed reaction products. a** The reaction product with the W43F mutant of crosslinked QhpC as the substrate for QhpG (*fragment d of Asp-N digestion). The calculated mass (*m/z*) is indicated within parentheses. **b** The reaction product formed by the wild-type (WT) and various mutants (E42A, R47A, R70A, W183F, and R314A) of QhpG (fragment d of Asp-N digestion).

carbonyl group hydrogen bonding to the N6 amino group of the ring. Several residues involved in binding of FAD, and those found in its vicinity are comparable to those of FAD-dependent halogenases[18–20]. In comparison with the SvCmlS structure, Ala14, Glu42, Arg89, Arg111, Asp260, and Pro267 of QhpG are fully conserved or conservatively replaced (Supplementary Fig. 8). In marked contrast to the well-conserved FAD-binding site, QhpG has considerably different secondary structural motifs in the region corresponding to the NAD(P)H-binding region of other flavin monooxygenases (FMOs). For example, an FMO from *Schizosaccharomyces pombe* (PDB ID: 2GV8)[24] has the NADPH-binding region consisting of a parallel five-stranded β-sheet, an antiparallel three-stranded β-sheet, and four α-helices, with a consensus sequence for the nucleotide-binding loop (GXSXXA), whereas the corresponding region of QhpG consists of an antiparallel five-stranded β-sheet and two α-helices without the consensus sequence (Supplementary Fig. 9). The small WH domain of QhpG resembles the domain often found in the core components of transcription systems as a DNA-binding motif[25]. The WHx domain is mounted over the catalytic domain that is linked by a random loop.

Two channels connecting the molecular surfaces to the *re-* and *si-*faces of the isoalloxazine ring of FAD have been identified. The narrow and deep *re-*face channel is open at the bottom of the large pocket between the two domains and composed of the main chains of Gly43–Val44 and Ser269–Asn271, and several hydrophobic residues (Val72, Trp74, Val174, Trp181, Trp183, Leu268, and Phe322) (Fig. 6c), most of which are conserved in SvCmlS and QhpG orthologs (Supplementary Fig. 8). In addition, several charged residues such as Arg70 and Glu84, also highly conserved in QhpG orthologs, are located at the rim of the channel. This channel corresponds to the substrate-binding site of flavoprotein monooxygenases[21,22], and the HOCl tunnel conserved in FAD-dependent halogenases[18–20]. It is most likely that the *re-*face channel plays as the binding site for the CTQ-precursor Trp43 in the substrate polypeptide (crosslinked QhpC) as described in the following section. The *si-*face channel connected to FAD is composed of a conserved residue (Glu42) and seven less conserved residues (Arg36, Phe38, Ala40, Arg89, Arg142, Gln143, and Arg240) (Fig. 6d and Supplementary Fig. 8). More hydrophilic residues than those of the *re-*face channel constitute the wide and shallow channel. On the *si-*face of FAD, in addition to Arg240, a loop of Glu34–Glu42 covers the bound FAD, and exhibits higher thermal factors than other regions. The side chains of Glu42 and Arg240 are apart by ~4.5 Å from each other, suggesting a weak electrostatic interaction, and are located over the middle of the isoalloxazine ring (Fig. 6d). It is possible that Glu42 and Arg240 act as a lid for the *si-*face channel, through which dissolved dioxygen is able to access to the isoalloxazine ring.

**Construction of docking models of binary and ternary complexes.** Based on the biochemical evidence showing the significant interaction between QhpG and crosslinked QhpC described above, we surveyed the surface area of QhpG that may interact with the crosslinked QhpC, and found a cluster of positively charged residues (Arg47, Arg70, Arg307, and Arg314) located near the entrance of the *re-*face channel (Supplementary Fig. 10a); the molecular surface of the γ-subunit of QHNDH is rich in acidic residues (Asp12, Asp33, Asp39, Asp56, Glu66, and Glu67)[9,10] (Supplementary Fig. 10b). Therefore, the crosslinked QhpC (substrate for QhpG) is assumed to interact electrostatically with the entrance of the *re-*face channel of QhpG. To validate this assumption, we first built a docking model using the crystal structure of γ-subunit in *Ps. putida* QHNDH[10] and the monomer structure (chain A) of QhpG. Among the top 10 complexes generated by the ZDOCK software and server[26], we selected the one that fits best with the above-described electrostatic interaction of γ-subunit with the large pocket between the two domains of QhpG (Supplementary Fig. 11a). We then manually built a structure model of crosslinked QhpC (without CTQ and the leader peptide) in the docking model of the QhpCG binary complex (Supplementary Fig. 11a, see also Supplementary Movie 1). In this docking model, the Asp39–Met51 loop of QhpC fit well into the deep *re-*face channel of QhpG, with Trp43 being placed close to the isoalloxazine ring of FAD (Supplementary Fig. 11b), in a manner similar to the modeled substrate L-kynurenine bound to kynurenine 3-monooxygenase[22]. Also, the side chain of the neighboring Trp42 was accommodated in a hydrophobic pocket formed by the conserved residues (Trp74, Leu268, and Phe322) of QhpG (Supplementary Fig. 11b). This docking model was further corroborated by site-directed mutagenesis of the residues located in the vicinity of the isoalloxazine ring of FAD (Glu42 and Trp183), or at the *re-*face channel entrance (Arg47, Arg70, and Arg314), which resulted in the nearly complete loss of the single-turnover activity of QhpG (Fig. 5b), and/or significant decreases in the affinity for the crosslinked QhpC (Supplementary Table 1).

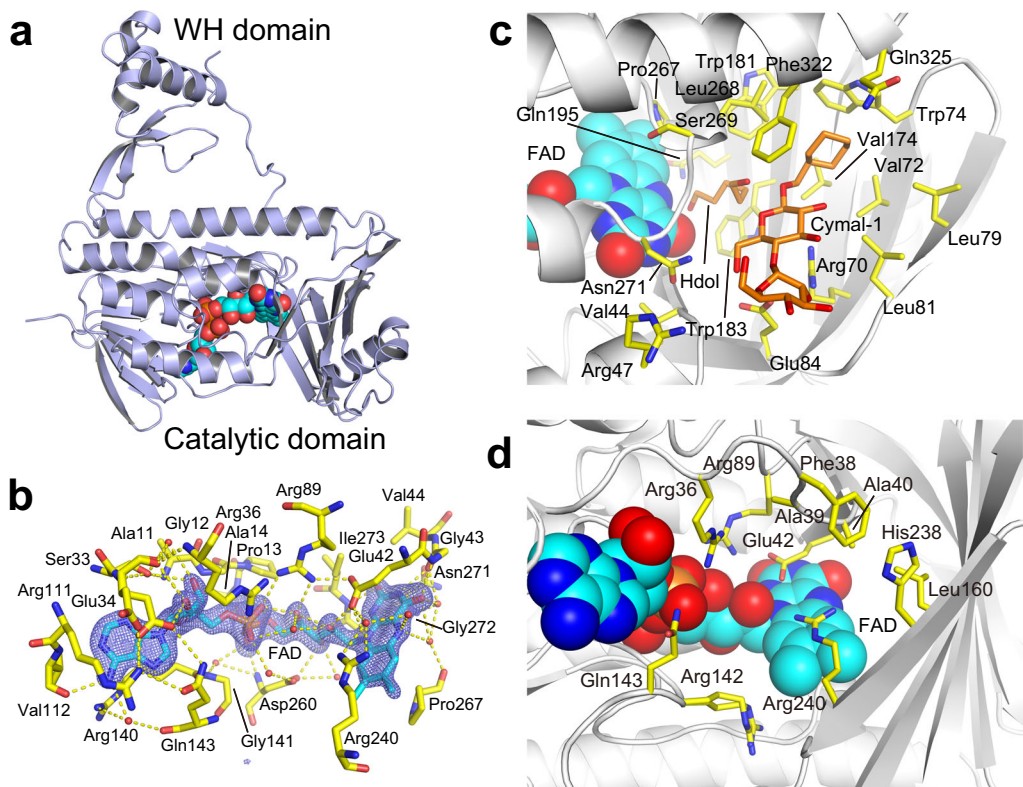

**Fig. 6 X-ray crystal structure of QhpG. a** Overall structure of QhpG monomer. The monomer structure (chain A) is depicted by a cartoon model with a spherical model of the bound FAD. **b** Structure of FAD-binding site. A stick model of the bound FAD is shown with the surrounding residues. An $F_o$–$F_c$ omit map for FAD, contoured at $5\sigma$, is depicted by blue mesh. Hydrogen bonds and water molecules are shown by yellow dotted lines and red small spheres, respectively. **c, d** Channels formed in the *re*-face (**c**) and *si*-face (**d**) sides of FAD (in chain B) is shown with stick models of the side chains (yellow) on the white cartoon model of QhpG. 1,6-Hexanediol (Hdol) and cyclohexyl-methyl-β-ᴅ-maltoside (Cymal-1) (additives in crystallization buffer) bound to the *re*-face channel are indicated by orange stick models. FAD is indicated by a spherical model.

For the construction of the docking model of the QhpCDG ternary complex, a structure model of QhpD had to be first generated using the SWISS-MODEL homology-modeling server (https://swissmodel.expasy.org)[27], because of the absence of its crystal structure. In the modeling, a radical SAM enzyme, a sactionine bond-forming enzyme CteB from *Clostridium thermocellum* (PDB ID: 5WGG)[28] was auto-selected as the template based on sequence homology. Subsequently, we could obtain a possible model of the QhpCDG ternary complex with ZDOCK by docking the above QhpD model structure to the model of the QhpCG binary complex (Supplementary Fig. 12). As reported previously for QhpD from *Pa. denitrificans*[14], the structure model of *Ps. putida* enzyme also has a large groove with sufficient space to accommodate the core QhpC polypeptide containing several negatively charged residues. The structure model of the ternary complex shows that QhpD binds the crosslinked QhpC at the large groove and from the opposite side of QhpC involved in the interaction with QhpG. Altogether, the crosslinked QhpC may be sandwiched by QhpD and QhpG and serves as the common protein substrate for these two enzymes, to undergo efficient and successive Cys–Asp/Glu crosslinking and Trp-dihydroxylation reactions.

## Discussion

The results described above reveal that QhpG is an atypical FAD-dependent oxygenase, catalyzing dihydroxylation of a peptidyl tryptophan. Such enzymes that regio- and stereo-specifically insert two or more hydroxyl groups into a single substrate have been reported for several cytochrome P450 monooxygenases[29];

5-epiaristolochene 1,3-dihydroxylase (EAH) involved in capsidiol biosynthesis in plant[30] (2× hydroxylation), DoxA[29] involved in doxorubicin biosynthesis (2× hydroxylation), Sky32[31] involved in the biosynthesis of a cyclic depsipeptide skyllamycin A (3× hydroxylation), TamI[29] involved in tirandamycin B biosynthesis (2× hydroxylation, 1 epoxidation), and MycG[29] involved in mycinamycin biosynthesis (1 hydroxylation, 1 epoxidation). A microbial nonheme $Fe^{II}$ α-ketoglutarate-dependent oxygenase (named OrfP) involved in antibiotic (streptothricin-F) biosynthesis[32] also inserts two hydroxyl groups into a single substrate. However, the enzyme that catalyzes dihydroxylation is so far unknown within the FMO family, although there is a related enzyme brominase (Bmp5)[33], which performs two successive regiospecific bromination reactions.

Referring to the mechanisms of EAH and OrfP, both accomplishing two independent and successive hydroxylation reactions within a single catalytic cycle, it is most likely that QhpG also catalyzes dihydroxylation in a successive manner. In the first hydroxylation step of the predicted reaction mechanism of QhpG (Fig. 7), the bound FAD is reduced to $FADH_2$ that reacts with dioxygen to form C4a-hydroperoxy-FAD, by analogy to other FAD-dependent monooxygenases[34,35] (step 1 → 2), which would then perform electrophilic substitution with the indole ring of Trp43 in QhpC (step 3), positioned nearly perpendicularly, close to the isoalloxazine ring of FAD (Supplementary Fig. 11b). We speculate that the indole ring C7 position is the first hydroxylation site because of elevated nucleophilicity of this position induced by the formation of a hydrogen bond between the indole ring N1 and FAD O4 atoms. Following the release of a water molecule, the 7-hydroxy-Trp residue is produced in QhpC, and

**Fig. 7 Predicted reaction mechanism of QhpG.** Details are described in the text (Discussion).

the oxidized FAD is regenerated in QhpG (step $4 \rightarrow 5$). In the second hydroxylation step, the formation of a new hydrogen bond between the 7-hydroxyl group of Trp43 and FAD O4 may bring a slight positional shift of the indole ring relative to FAD (step 6). A small conformational change of the Asp39–Met51 loop of QhpC (Supplementary Fig. 11b) may also be induced after the first hydroxylation. The remaining dithionite again reduces FAD to $FADH_2$, which then reacts with dioxygen to form C4a-hydroperoxy-FAD again, for insertion of the second hydroxyl group at the C6 position of 7-hydroxy-Trp, finally yielding the 6,7-dihydroxy-Trp43 (steps $6 \rightarrow 7 \rightarrow 8$). The predicted reaction mechanism is consistent with those proposed for EAH[30] and OrfP[32] in that the first mono-hydroxylated intermediate re-engages in the reaction without being released from the catalytic center. The observation that the crosslinked QhpC with two oxygen atoms incorporated was the major product of the QhpG reaction (Fig. 3a), suggesting the faster reaction rate of QhpG for the mono-hydroxylated intermediate (7-hydroxy-Trp) than for the initial substrate (crosslinked QhpC), is in agreement with the reaction catalyzed by EAH[30].

The positive effect of the QhpCDG ternary complex formation exerted on both of the QhpD-catalyzed thioether bond formation (Fig. 2a) and the QhpG-catalyzed dihydroxylation of the Trp residue (Fig. 3a) may be attributed to structural stabilization of the common substrate, that is, the QhpC polypeptide chain, composed mostly of random coils with only two short α-helices (Supplementary Fig. 11b). It is assumed that the QhpG protein facilitates the QhpD reaction by capturing the partially crosslinked QhpC, and vice versa, the QhpD protein helps the QhpG reaction by stably holding the protein substrate (crosslinked QhpC) through the N-terminal leader peptide[14]. Supporting the structural significance, the QhpG reaction does not occur in the absence of QhpD (Fig. 3a), even though the bound FAD is reduced by sodium dithionite in the presence of crosslinked QhpC (Supplementary Fig. 2b). The physiological electron donor for the QhpG reaction is unknown at present. An electron-transfer protein existing in bacterial cells (e.g., ferredoxin, flavodoxin, and thioredoxin) may directly supply electrons for QhpG. Another possibility is that electrons supplied by an electron-transfer protein may be transferred to QhpG via QhpD that contains [4Fe–4S] clusters (one RS and two auxiliary clusters)[13,14] within the QhpCDG ternary complex. These possibilities remain to be examined in future studies.

The role played by QhpG in the quinone cofactor biogenesis is worth comparing with those of the modifying enzymes involved in other tryptophylquinone-generating systems[6], such as MauG[4,36,37], a di-heme protein participating in the biosynthesis of tryptophan tryptophylquinone (TTQ), the first tryptophylquinone cofactor discovered in methylamine dehydrogenase (MADH)[38], and LodB[39] and GoxB[40–42], flavoproteins required for the formation of CTQ identified in l-lysine ε-oxidase (LodA)[43] and glycine oxidase (GoxA)[44], respectively. Most importantly, the target Trp residue in the substrate proteins for MauG, LodB, and GoxB is a mono-(7-)hydroxy-Trp[4,36,37], whereas that for QhpG is an unmodified Trp. Thus, QhpG inserts both oxygen atoms into the Trp of CTQ, whereas in the biogenesis of TTQ in MADH and CTQ in LodA and GoxA, the substrate for the modifying enzyme has the first hydroxyl present and only the second is added along with the formation of the Trp–Trp or Trp–Cys crosslink. Formation of the initial mono-hydroxy-Trp intermediate in MADH and LodA/GoxA is thought to be an autocatalytic process[36,45], which appears to be copper-ion dependent in LodA[45], with the participation of an Asp residue[40,46,47] located close to the cofactor, and strictly conserved in all tryptophylquinone enzymes. However, a recent study on GoxA has shown that mutation of the corresponding Asp678 does not abolish CTQ formation[48]. In addition, the corresponding Asp33 in QhpC may be placed in the equivalent position only after the formation of the CTQ thioether bond (Supplementary Fig. 11b), apparently excluding its role in CTQ biogenesis, although a catalytic role in amine oxidation has been suggested[49]. Both QhpG (crystal) and GoxB (model)[42] show the highest structural similarity to alkylhalidase CmlS[20] with FAD bound to a nearly equivalent position and in an almost identical conformation (Supplementary Fig. 7c). However, FAD is bound loosely in GoxB[42], but very tightly in QhpG, suggesting that QhpG belongs to the category of single-component monooxygenases[34], but without bound $NADP^+$ or using NADPH as a co-substrate. The crystal or modeled structure of MauG-preMADH[50] and GoxB-GoxA[42] complexes shows the catalytic centers (di-heme in MauG, FAD in GoxB model) being far away from the target Trp residue in their partner protein substrates, indicating long-range electron transfer for remote Trp modification. In clear contrast, the CTQ-precursor Trp in the crosslinked QhpC substrate can be placed close to the bound FAD of QhpG, as shown in the docking model (Supplementary Fig. 11a), and may directly undergo hydroxylation by C4a-hydroperoxy-FAD (Fig. 7). The γ-subunit

(mature QhpC) of QHNDH contains three Cys-to-Asp/Glu thioether bonds, instead of six disulfide bonds contained in the TTQ-bearing β-subunit of MADH[4,47]. Finally, the multistep TTQ biogenesis in MADH occurs after the TTQ-less β-subunit is translocated into the periplasm[51,52], whereas all the single-turnover reactions by QhpD, QhpG, and QhpE on QhpC occur within the less aerobic cytoplasm[11]. By analogy with TTQ biogenesis in MADH, the final oxidation to CTQ of 6,7-dihydroxy-Trp in the crosslinked QhpC (pre-γ-subunit) produced by the QhpG reaction may be catalyzed by the two *c*-type hemes that are contained within the α-subunit, presumably after the periplasmic translocation of both of the pre-γ- and α-subunits[11]. Altogether, it is now evident that markedly divergent strategies have evolved to convergently generate the tryptophylquinone cofactor in different proteins.

In conclusion, the γ-subunit intermediate of QHNDH containing the 6,7-dihydroxy-Trp and triple thioether crosslinks is produced in vitro from the nascent linear polypeptide of QhpC by the collaborative activities of two unusual modifying enzymes, QhpD and QhpG. The single-turnover feature of the reactions of QhpD[14], QhpE[15], and QhpG (this paper) is consistent with the fact that the genes for these proteins are encoded within the same operon (*qhp*) as their substrate (QhpC) and they are expressed altogether under the control of the *n*-butylamine inducible transcriptional regulator[11]. Thus, a single use of each modifying enzyme is allowed in processing a single molecule of the substrate polypeptide, as reported for the ribosomally synthesized and posttranslationally modified peptides with various biological activities[53].

## Methods

**Plasmid construction.** Expression plasmids for QhpG, QhpC, and QhpD from *Ps. putida* IFO 15366 (NBRC 15366) were constructed using either an *E. coli* expression vector pET-15b or a broad-host-range vector pBBR1, mostly according to the standard molecular genetic protocols (Supplementary Fig. 13). Coding regions of these proteins were amplified by PCR using sense primers containing an *Nde*I site at the 5′ terminus and antisense primers containing a *Bam*HI or *Nhe*I site at the 3′ terminus (QhpC (+), QhpC (−) for QhpC; QhpD (+), QhpD(−) for QhpD; and QhpG (+), QhpG (−) for QhpG; Supplementary Table 3) with *Ps. putida* genome DNA. For plasmid construction of QhpG, the C-terminal Ala430 was replaced by Val in response to the designed primer sequence. A His6-tag and a Tobacco etch virus (TEV) protease digestion site (GSSHHHHHHDY-DIPTTENLYFQG) were appended to the N terminus of QhpG (pET-His6-TEV-QhpG) by inserting a DNA fragment obtained from annealed synthetic oligonucleotides (His6-TEV (+), His6-TEV (−); Supplementary Table 3). The constructs pET-His6-QhpD and pBBR-QhpC-St2, in which a His6-tag and an St2-tag were appended to the N terminus of QhpD and the C terminus of QhpC, respectively, were prepared by inserting the PCR-amplified fragments into the expression plasmids for QhpC and QhpD from *Pa. denitrificans* Pd1222 constructed in the previous work[14]. The expression plasmid for QhpA from *Pa. denitrificans* (pRK-His6-QhpA) was constructed by PCR amplification with synthetic oligonucleotides (pdQhpA (+), pdQhpA (−); Supplementary Table 3)[11]. A His6-tag (GSSHHHHHHSSG) was appended at the C terminus of the signal peptide in the construct by PCR amplification using synthetic oligonucleotides (His6-pdQhpA (+), His6-pdQhpA (−); Supplementary Table 3). The plasmids for site-directed mutagenesis of QhpG and QhpC (Supplementary Fig. 13) were prepared by the PCR-based method using primers listed in Supplementary Table 3 (QhpG-E42A (+), QhpG-E42A (−) for QhpG-E42A mutant; QhpG-R47A (+), QhpG-R47A (−) for QhpG-R47A mutant; QhpG-R70A (+), QhpG-R70A (−) for QhpG-R70A mutant; QhpG-W183F (+), QhpG-W183F (−) for QhpG-W183F mutant; QhpG-R314A (+), QhpG-R314A (−) for QhpG-R314A mutant; and QhpC-W43F (+), QhpC-W43F (−) for QhpC-W43F mutant). The absence of PCR-derived errors in all constructs was confirmed by sequencing the entire coding regions.

**Protein expression and purification.** The inactive QHNDH produced in the periplasm of the Δ*qhpG* mutant strain of *Pa. denitrificans* was purified from the cells transformed with pRK-His6-QhpA (Supplementary Fig. 13)[11,13]. Briefly, the periplasmic fractions of the cells were added with 0.375 mM MgCl2, 500 mM NaCl, and 10 mM imidazole and centrifuged at $20,000 \times g$ for 30 min. The supernatant solution was loaded at 1 ml min⁻¹ onto a 5-ml HisTrap HP column (GE Healthcare) pre-equilibrated with 20 mM Tris-HCl, pH 7.4, and 500 mM NaCl (buffer A) containing 10 mM imidazole. The column was washed with buffer A containing 10 mM imidazole and the protein was eluted with a linear gradient of

10–250 mM imidazole in buffer A. Fractions containing QHNDH was pooled and stored at 4 °C until use.

The QhpCD binary complex was expressed in *E. coli* C41 (DE3) cells carrying expression plasmids pBBR-QhpC-St2 and pET-His6-QhpD (either from *Pa. denitrificans* or *Ps. putida*) and purified using the HisTrap and Strep trap affinity columns[14]. After chemical reconstitution of the [Fe–S] clusters of QhpD in the as-purified QhpCD binary complex[14], the complex (~180 μM) dissolved in 25 mM Tris-HCl, pH 8.0, containing 150 mM NaCl and 10% (w/v) glycerol was anaerobically incubated at ~20 °C for ~15 h with 1 mM sodium dithionite and 1 mM SAM, in the absence or presence of ~180 μM purified QhpD (with QhpD derived from *Pa. denitirificans* or *Ps. putida*, respectively), for the QhpD-catalyzed intra-peptidyl thioether bond formation in QhpC. The free linear and crosslinked QhpC proteins were prepared before and after the QhpD-catalyzed reaction, respectively, by removing the QhpD protein by heat denaturation at 60 °C for 20 min and following centrifugation.

To prepare the wild-type or mutant protein of QhpG, *E. coli* C41 (DE3) cells carrying a relevant expression plasmid were grown at 37 °C for 3 h by reciprocal shaking at 160 r.p.m. and further at 25 °C and 180 r.p.m. for 20 h in an Overnight Express™ Instant TB Medium (Novagen) supplemented with ampicillin (50 μg ml⁻¹). The cells were harvested by centrifugation at $5000 \times g$ for 10 min and stored at −80 °C until use. For purification of QhpG, the frozen cells were suspended in 50 mM sodium phosphate buffer, pH 8.0, containing 300 mM NaCl and 10% glycerol (v/v) (buffer B) plus 5 mM imidazole, 1 mM FAD, and cOmplete Protease Inhibitor (Roche) (1 tablet per 50 ml of cell suspension), and disrupted by ultrasonic oscillation. The cell-free extract was obtained by centrifugation at $20,000 \times g$ for 20 min, and was applied to a Ni-Sepharose FF column (5-ml bed volume, GE Healthcare) pre-equilibrated with buffer B. After washing the column with 100 ml of buffer B containing 5 mM imidazole, QhpG was eluted in a stepwise manner with buffer B containing 10, 20, 50, 100, and 200 mM imidazole. The fractions colored yellow (eluted by 50–100 mM imidazole) were collected, and 1 mM FAD was supplemented. After twice repeated dialysis against buffer B, the protein solution was further applied to a HisTrap HP column (5-ml bed volume, GE Healthcare) pre-equilibrated with buffer B containing 5 mM imidazole. After extensive washing with buffer B containing 5 mM imidazole, QhpG was eluted with a linear gradient of 5–100 mM imidazole in buffer B. The fractions containing QhpG were pooled, added with 1 mM FAD, dialyzed against buffer B, and concentrated to ~10 mg ml⁻¹. To remove the N-terminal His6-tag, the concentrated protein was digested with a His6-tagged TEV protease[54] (added at 1% protein weight) for 48 h at 6 °C. The His6-tagged TEV protease was produced by an *E. coli* expression system with plasmid pRK793, which was a gift from David Waugh (Addgene plasmid #8827; http://n2t.net/addgene:8827; RRID:Addgene_8827) and purified according to the Addgene protocol. After the digestion, the protease and released His6-tag were removed by repeating the HisTrap HP purification, and the tag-less QhpG was collected from fractions containing 40–50 mM imidazole based on its weak affinity for the column. After addition of 1 mM FAD, the protein solution was dialyzed against buffer B. Finally, the purified QhpG was concentrated to >10 mg ml⁻¹ and stored at −80 °C in the presence of 1 mM FAD. Excess FAD was removed by a PD-10 column (GE Healthcare) before use.

**Characterization of QhpG.** The protein concentration of purified QhpG was determined with a molecular extinction coefficient at 280 nm ($\varepsilon_{280} = 71{,}960$ M⁻¹ cm⁻¹) calculated from the amino acid sequence of QhpG with a program ProtParam[55], by subtracting the contribution of FAD absorption ($\varepsilon_{280} = 2340$ M⁻¹ cm⁻¹) measured in the same buffer. The FAD content was determined spectrophotometrically in buffer B containing 8 M urea with a molecular extinction coefficient at 450 nm ($\varepsilon_{450} = 11{,}300$ M⁻¹ cm⁻¹)[56]. The approximate molecular size of QhpG in solution was determined by a gel-filtration method with the purified QhpG (6 mg ml⁻¹) applied onto a Superdex 200 10/300 GL column using 50 mM sodium phosphate, pH 7.0, containing 150 mM NaCl as an eluent at a flow rate of 0.5 ml min⁻¹. Chymotrypsinogen A (MW, 25,000), ovalbumin (43,000), albumin (67,000), aldolase (158,000), catalase (232,000), and ferritin (440,000) were used for standards.

**MALDI-TOF MS analysis.** The purified QHNDH from the Δ*qhpG* mutant strain of *Pa. denitrificans* was denatured by the addition of 10% (w/v) trichloroacetic acid and washed twice with cold acetone. The precipitates collected by centrifugation were thoroughly dried, then dissolved in 50 μl of 6 M urea in 50 mM potassium phosphate, pH 7.5, containing 1 mM *tris*(2-carboxyethyl)phosphine (TCEP), and incubated at 37 °C for 1 h[11,14]. For chemical modification of the free sulfhydryl groups, a 10-μl aliquot of the solution was mixed with 1 μl of 500 mM IAA in 50 mM potassium phosphate, pH 7.5, and the mixture was kept at room temperature for 1 h. The mixture was acidified with 2% (v/v) formic acid, adsorbed in a desalting C18 ZipTip pipette tip (Millipore), and eluted with 10 μl of 0.1% (v/v) trifluoroacetic acid (TFA) in 50% (v/v) acetonitrile. A 1-μl aliquot of the eluate was analyzed with a Bruker Ultraflex III MALDI-TOF mass spectrometer using 1 mg ml⁻¹ sinapic acid (Bruker) dissolved in 90% (v/v) acetonitrile containing 0.1% (v/v) TFA as a matrix, which was co-crystallized with the protein by the drying-droplet method. Before each MS analysis, mass calibration was done using Protein Calibration Standard I (Bruker).

For identification of the bound cofactor in QhpG, the purified protein (~0.125 mg in 10 μl distilled water) was incubated at 50 °C for 10 min and the precipitated protein

was removed by centrifugation at 4 °C. A 1-µl aliquot of the supernatant was subjected to MALDI-TOF MS analysis using a nearly saturated concentration of 2,5-dihydroxy benzoic acid dissolved in 50% (v/v) acetonitrile and 2.5% (v/v) formic acid as a matrix. FAD standard was dissolved at 1 mg ml$^{-1}$ in distilled water. The mass spectra were calibrated with Peptide Calibration Standard II (Bruker).

**Mobility shift assay on native PAGE.** Protein samples dissolved in 8 µl of 25 mM Tris-HCl, pH 8.0, containing 150 mM NaCl and 12.5% (w/v) glycerol were analyzed by native PAGE using an Any kD Mini-PROTEAN TGX precast gel (Bio-Rad) according to the manufacturer's protocol. Pre-running at a constant electric current of 34 mA was done for 15 min. After applying the samples, electrophoresis (10 mA, 40 V) was run at 4 °C for ~3 h using a tris-glycine buffer system (25 mM Tris, 192 mM glycine, pH 8.3). In the analysis of the QhpCDG ternary complex formation, 1 mM sodium dithionite was added to the electrophoresis buffer to remove dissolved dioxygen in the gel during the pre-running, and all procedures were done under anaerobic conditions maintained in a glovebox filled with 99.999% (v/v) nitrogen gas; oxygen concentration was routinely maintained below 0.1% (v/v) by monitoring with an oxygen sensor.

**BLI assay.** Protein–protein interaction was analyzed quantitatively by BLI assays conducted in 10 mM Tris-HCl, pH 7.4, containing 150 mM NaCl, 0.1% (w/v) bovine serum albumin, and 0.02% (v/v) Tween 20 (kinetics buffer) at 25 °C using a BLItz System (Molecular Devices, LLC., CA). Streptavidin-coated biosensors were hydrolyzed for 30 min in 250-µl kinetics buffer. A baseline was measured with a ligand-free biosensor in the kinetics buffer for 30 s prior to the immobilization step. A 4-µl aliquot of 0.5 µM protein solution (linear or crosslinked QhpC) was applied for immobilization through the St$_2$-tag to the surface of biosensor tips for 4 min and the baseline was measured again with the ligand-attached biosensor in the kinetics buffer for 6.5 min. An association step was monitored by applying an analyte solution (4 µl of 0.06–0.5 µM QhpG) to the biosensors for 4 min. Subsequently, a dissociation step was monitored by pouring the kinetics buffer (250 µl) for 4 min. To correct for the nonspecific binding, reference data with the same concentrations of analyte were also measured with a ligand-free biosensor. To determine the dissociation constant ($K_D$), association rate constant ($k_a$), and dissociation rate constant ($k_d$), the binding data obtained were analyzed by a BLItz Pro1.2 software (Molecular Devices, LLC, CA) using a global analysis mode with corrections for association and dissociation steps.

**Assay of QhpG activity.** After conducting the QhpD-catalyzed thioether bond formation as described above, a 250-µl reaction mixture of the QhpCDG ternary complex (52 µM) in 25 mM Tris-HCl, pH 8.0, containing 150 mM NaCl and 10% (w/v) glycerol (buffer C) was incubated with 3 mM sodium dithionite for 1 h at room temperature in the glovebox. The reduced mixture was taken out from the glovebox, added with twice volumes of O$_2$-saturated buffer C, and further incubated for 1 h at room temperature under atmospheric conditions. For the reaction in the absence of QhpD, the free crosslinked QhpC (18 µM) in buffer C was mixed with 22.5 µM QhpG, reduced with 1 mM sodium dithionite in the glovebox, and further incubated with twice volumes of O$_2$-saturated buffer C, as described above. The reaction product was precipitated by the addition of cold acetone and was subjected to sodium dodecyl sulfate-PAGE followed by blotting onto a polyvinylidene difluoride membrane (Immobilon-PSQ) and quinone staining with 2 M potassium glycinate, pH 10.0, containing 0.24 mM nitro blue tetrazolium[13]. Another portion of the precipitated reaction product was used for digestion with endoproteinase Asp-N (Sigma-Aldrich) (added at 1% protein weight) according to the manufacturer's protocol. The Asp-N digests were desalted with a C$_{18}$ ZipTip pipette tip and subjected to the MALDI-TOF MS analysis as described above using 2,5-dihydroxy benzoic acid dissolved in 90% acetonitrile and 0.01% TFA as a matrix. A collision-induced dissociation method with Ar gas was used for peptide fragmentation in the MS/MS analysis. For the reaction in H$_2$$^{18}$O, buffer C dissolving QhpG and the QhpCD binary complex was exchanged with H$_2$$^{18}$O (≥98 atom% $^{18}$O; Taiyo Nissan Co., Tokyo, Japan) by repeating concentration and dilution three times using an Amicon 10 K concentrator. The $^{16}$O$_2$-saturated buffer C was also prepared from the H$_2$$^{18}$O buffer C. The QhpG reaction and MS analysis of the product were conducted as described above.

**Crystallization and data collection.** The buffer used for the storage of purified QhpG was exchanged with a freshly prepared buffer consisting of 10 mM Tris-HCl, pH 8.0, 0.1 M NaCl, and 0.1 mM dithiothreitol using a PD-10 column. The protein solution was concentrated to 12.5 mg ml$^{-1}$, filtrated through a 0.22-µm centrifugal filter (Millipore), and used for crystallization screening. Initial crystallization conditions were screened with commercially available screening kits by the sitting-drop vapor diffusion method. After optimization, the crystals used for data collection were obtained by the sitting-drop method at 4 °C in the reservoir solution consisting of 100 mM HEPES (pH 7.0), 20 mM MgCl$_2$, 17% (v/v) poly(acrylic acid sodium salt) 5100, 1.8% (w/v) 1,6-hexanediol, and 34 mM cyclohexyl-methyl-β-D-maltoside. Thin platy crystals (ca. 0.2–0.3 mm size) appeared within 1–2 months. After cryoprotection with the crystallization buffer containing 30% (v/v) glycerol, the crystals were mounted on thin nylon loops (ϕ, 0.2–0.3 mm) and frozen by flash

cooling at 100 K in a cold N$_2$ gas stream or in liquid nitrogen. Hg derivatives were prepared by soaking the native crystals in the crystallization buffer containing 30% (v/v) glycerol and a saturated concentration (ca. 5 mM) of carbon tetra(acetoxymercuride) for overnight. Diffraction data sets were collected with a synchrotron X-radiation at SPring-8 (Hyogo, Japan) at 100 K in the beamline station BL44XU using a CCD detector EIGER X 16 M (Dectris) at $\lambda = 0.900$ Å for a native crystal and using a CCD detector MX300HE (Rayonix) at $\lambda = 1.0070$ Å for an Hg-derivatized crystal. The collected data were processed, merged, and scaled using the program XDS[57] or HKL2000[58]. The space group was $P2_1$ with the unit cell dimensions of $a = 88.5$ Å, $b = 51.8$ Å, and $c = 101.8$ Å, $\beta = 99.8°$ in the native crystal. The cell content analysis suggested that two molecules are in the asymmetric unit. The details and statistics of the data collection are summarized in Supplementary Table 2.

**Structure determination and refinement.** The structure of QhpG was solved by single isomorphous replacement with anomalous scattering of the Hg-derivatized crystal. The positions of two Hg sites were determined by calculating anomalous peaks with the programs SHELXC/D[59] within the program suite autoSHARP[52]. Initial phases were calculated using the program SHARP[60], and subsequent cycles of density modification and automatic model building with autoSHARP[60] gave an initial model containing 55% of the peptide backbone in the asymmetric unit. NCS relating two QhpG molecules in the asymmetric unit was manually determined by analyzing the electron density along with the positions of the two Hg sites and their neighboring identical α-helices in the initial model. The resulting NCS matrices together with the phases and coordinates of the autoSHARP output was subsequently input to the program PHENIX AutoBuild[61] for further cycles of density modification and automatic model building, which gave an overall figure of merit of 0.70 and built 70% of the peptide backbone in the asymmetric unit including 58% of the side chains. The resulting map was of good quality and clearly exhibited electron density corresponding to two FAD molecules. The remaining model was built manually into the map using the program Coot[62]. Refinement using the program PHENIX[61], and manual model rebuilding cycles produced the final model. Analysis of the stereochemistry showed that the model was of good quality, with >99.8% of the residues falling in the allowed regions. One proline residue (Pro342) in each QhpG monomer was found in the *cis*-conformation. PyMOL version 1.8 or 2.4 (Schrödinger Inc. New York, USA) was used for figure drawings. The refinement statistics are summarized in Supplementary Table 2, and the coordinates and structure factors have been deposited in the PDB (PDB entry ID: 7CTQ).

**Construction of docking models.** The initial docking model of the QhpCG binary complex was built with the ZDOCK software and server[26] using the crystal structure of γ-subunit in *Ps. putida* QHNDH (PDB entry ID: 1JMX)[10] as ligand and the monomer structure (chain A) of QhpG as the receptor. Starting from this initial model, Coot[62]-based manual modeling was carried out as follows: The thioether bond of CTQ was cleaved and the Trp42–Trp43 dipeptide portion was separately placed at the bottom of the *re*-face channel of QhpG so that the indole ring of Trp43 overlaps with the phenyl ring of L-kynurenine modeled in the active site of kynurenine 3-monooxygenase[22]. The Asp39–Met51 loop lacking Trp42–Trp43 was moved into the *re*-face channel and re-connected to the Trp42–Trp43 portion. The docking model of the QhpCG binary complex thus constructed was energy-minimized with a Schrödinger program suit (Schrödinger Inc., New York, USA). For model building of the QhpCDG ternary complex, a structure model of QhpD was first generated using the SWISS-MODEL homology-modelling server (https://swissmodel.expasy.org)[27]. Template molecules for modeling were searched with BLAST[63] and HHblits[64] against the SWISS-MODEL template library, which includes the PDB. Finally, the model of the QhpCDG ternary complex was built with the ZDOCK software and server[26] using the modeled QhpD structure as ligand and the QhpCG binary complex as the receptor.

**Reporting summary.** Further information on research design is available in the Nature Research Reporting Summary linked to this article.

## Data availability
Relevant data are included in this article and its Supplementary information. Atomic coordinates and structural factors have been deposited in the Protein Data Bank (accession code 7CTQ). The structure of the mature form of QhpC (γ-subunit) was extracted from the previously published structure of QHNDH (1JMX). Atomic coordinates of an FMO (PDB ID: 2GV8) and an FAD-dependent halogenase (PDB ID: 3I3L) were used for structural comparison, and that of a radical SAM enzyme, CteB (PDB ID: 5WGG) was for homology modeling of QhpD. Other data supporting the findings of this manuscript are available from the corresponding author upon reasonable request. Source data are provided with this paper.

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

## Acknowledgements

We thank the staff of the Comprehensive Analysis Center, ISIR, Osaka University for technical assistance in MALDI-TOF MS analyses. This work was performed using synchrotron beamline stations BL44XU at SPring-8, under the Cooperative Research Program of the Institute for Protein Research, Osaka University (Proposal Nos.: 2014A6912, 2014B6912, 2015A6508, 2016A6608, 2017B6709, 2018A6807, 2018B6807, 2019A6907, and 2019B6907). This research was supported by JSPS KAKENHI Grant Numbers 18J22104 to T.Oo, JP 23570135 and JP 16K07691 to T.N., and JP 24658288 and JP 15K07391 to T.Ok., the Operational Program Education for Competitiveness, and funding from the Network Joint Research Center for Materials and Devices.

## Author contributions

T.Oo., T.N., K.T., S.K., K.Kob., and T.Ok. participated in the research design. T.Oo., T.N., and T.Ok. conducted X-ray crystal analysis. T.Oo., T.N., K.Koz, and T.Ok. purified proteins, conducted MS analysis, and assayed QhpG activity. T.Oo. and K.O. performed BLI assays. T.Oo., T.N., K.Koz, K.O., S.K., K.Kob., K.T., and T.Ok. performed data analysis, and wrote or revised the manuscript.

## Competing interests

The authors declare no competing interests.
