## [Peer Review File · Nature Communications]

REVIEWER COMMENTS

Reviewer #1 (Remarks to the Author):

This is an outstanding paper. It employs a wide range of approaches and challenging experiments to answer a longstanding and important question regarding the mechanism of CTQ biosynthesis in this enzyme. These results are relevant to the broad topic of unusual mechanisms of post-translational modifications to form protein-derived cofactors. It also describes a novel catalytic role for a flavoprotein. The evolutionary implications regarding the mechanisms tryptophylquinone biosynthesis in different enzymes and different organisms is also interesting and well stated.

I have a few comments.

1. Can the authors speculate on what might be the natural electron donor for the reaction, since it is not NADPH and dithionite was used in the study?
2. The reaction that was achieved is referred to as a single turnover. Does that mean that one molecule of enzyme could only form CTQ on one molecule of substrate? Was it possible to observe multiple turnovers with excess protein-complex substrate? If not why?
3. The evidence of CTQ formation was primarily based on mass spec analysis. It is convincing. Did the product with CTQ exhibit an absorbance spectrum with the characteristic CTQ spectral features? Was it possible to combine the gamma subunit product with the alpha and beta subunits and demonstrate catalytic activity?
4. The authors refer to the structure of GoxB in ref. 35 and correctly point out some distinctions. A somewhat remarkable point, which I do not think that they made, is that the GoxB structure was a homology model that used the structure of the Alkylhalidase CmlS. It received the best QMEAN Z score from a screen of the PDB database. This is the same protein (ref 18) that showed the highest structural similarity to QhpG. I find it amazing that both GoxB and QhpG are most similar to this halidase.
5. Another point that I think is very interesting and should be given more emphasis is that QhpG inserts both oxygens into the Trp of CTQ. I believe that in all other known cases, TTQ in methylamine dehydrogenase, and CTQ in LodA and GoxA, the substrate for the modifying enzyme has the first hydroxyl present and only the second is added along with the formation of the crosslink.

Reviewer #2 (Remarks to the Author):

This manuscript by the Okajima group reports functional and structural analysis of a rather special flavoprotein (NADP) monooxygenase responsible for biosynthesis of cysteine tryptophylquinone (CTQ). Intriguingly, CTQ is generated from a protein-derived quinone cofactor.

The work reported here is comprehensive. It convincingly demonstrates the enzymatic activity of the flavoprotein. The high-resolution structure of QHNDH is well determined and informative. Functional analysis is sound and supports the main conclusion.

There are a few relatively minor concerns:

Abstract: "in the protein substrate containing triple intra-peptidyl sulfur-to-methylene carbon thioether bonds formed by a radical S-adenosylmethionine enzyme."

Awkward phrasing

Line 207-208: "In view of the less extensive interactions 207 between the two monomers, the dimerization appeared to be a crystallographic artifact."

This conclusion should be more vigorously tested using such as Pissa analysis and even enzymatic experiment (Hills coefficient).

Ramachandran Outlier 0.2% - This is not insignificant for protein of this size and structure of this resolution. Authors should examine these outliers and see whether they have any potential connection to amino acid modifications.

Lines 85-86: "three intra-peptidyl thioether bonds 86 formed between Cys and Asp or Glu residues"

I'm not sure I understand this. Formation of a thioether from Cys and Glu/Asp (See Figure 1B) involves the loss of 2 x H, or 2 Da, so three thioether bonds would involve a loss of 6 Da, not 28.

Line 168, "m/z, 2054.673 ± 0.213"

significant figures, should be to one decimal place.

Lines 287-288: "regiospecifically insert two hydroxyl groups into a single substrate,"

Could have better context. There are a few more examples of enzymes that perform multiple, regio- and stereospecific oxidation reactions.

Some P450 examples:

Sky26, a monooxygenase that performs 3 regiospecific hydroxylations: Pohle, S., Appelt, C., Roux, M., Fiedler, H.-P., and Süssmuth, R. D. (2011) Biosynthetic Gene Cluster of the Non-ribosomally Synthesized Cyclodepsipeptide Skyllamycin: Deciphering Unprecedented Ways of Unusual Hydroxylation Reactions. 133, 6194–6205

DoxA, doxorubicin biosynthesis (2x hydroxylation)

MycG, mycinamycin biosynthesis (1 hydroxylation, 1 epoxidation)

TamI, tirandamycin B biosynthesis (2 x hydroxylation, 1 epoxidation).

(above 3 examples cited in review by Greule, A., Stok, J. E., De Voss, J. J., and Cryle, M. J. (2018) Unrivalled diversity: the many roles and reactions of bacterial cytochromes P450 in secondary metabolism. Nat. Prod. Rep. 35, 757–791)

Also related is the flavin dependent brominase Bmp5, which performs two successive regiospecific bromination reactions:

1. Agarwal, V., Gamal, El, A. A., Yamanaka, K., Poth, D., Kersten, R. D., Schorn, M., Allen, E. E., and Moore, B. S. (2014) Biosynthesis of polybrominated aromatic organic compounds by marine bacteria. Nat Chem Biol. 10, 640–647

Lines 299-301: "We speculate that the indole ring C7 position is the first hydroxylation site because of elevated electronegativity by the formation of a hydrogen bond between the indole ring N1 and FAD O4 atoms."

Replace "...elevated electronegativity....." with "elevated nucleophilicity of this position induced...."

Lines 365-368: "The protein machinery consisting of the two enzymes swallows a raw material (QhpC) and crafts a decorated product with a complicated structure (pre-g-subunit), like magician's hands manipulating a linear rope and changing it into a multi-knotted lariat with a pendant."

Purple prose. I would delete this. The authors should note that this is very analogous to the biosynthesis of RiPPs.

Reviewer #3 (Remarks to the Author):

Manuscript: Functional and structural characterization of a flavoprotein monooxygenase essential for biogenesis of tryptophylquinone cofactor

Authors: Toshinori Oozeki, Tadashi Nakai, Kazuki Kozakai, Kazuki Okamoto, Shun'ichi Kuroda, Kazuo Kobayashi, Katsuyuki Tanizawa and Toshihide Okajima^{1*}

This manuscript reports the first structural and functional characterization of QhpG as a catalyst for the biogenesis of CTQ within QhpC, one of three subunits that constitute the mature, multi-subunit enzyme quinohemoprotein amine dehydrogenase (QHNDH). In the past, the hydroxylation of Trp in QhpC was proposed to occur via a flavin-dependent monooxygenase (FMO) based on sequence similarity. Using a combination of biochemical, biophysical and structural biology methods, the authors in this manuscript discover that the qhpG gene encoded protein that uses one equivalent of FAD to conduct dihydroxylation on the Trp residue within QhpC, in the presence of a reducing reagent and subsequent to the formation of interchain thioether linkages catalyzed by QhpD. The interaction between substrate QhpC and QhpG before and after modification by QhpD suggests that the QhpG modifies the crosslinked QhpC. The authors have obtained a fairly high-resolution X-ray crystal structure of QhpG with FAD bound in the active site, and their computational docking suggests that the active site can accommodate the substrate QhpC possibly via electrostatic interaction.

Although FMO has been reported in numerous biosynthetic pathways, this is the first study that suggests that FMO modifies the Trp in the maturation of CTQ cofactor in a dihydroxylation manner. If the mechanism as proposed is correct, it is rare for FMO. However, other than the demonstration of a dihydroxy-Trp product, there are relatively few data that can inform the mechanism presented by the authors. Three key issues need to be addressed, before the paper can be recommended for publication.

First, the paper would be considerably strengthened by experiments that can distinguish whether the two oxygens in the modified Trp come from O₂ vs H₂O (via the addition of 18O₂-saturated buffer or H₂¹⁸O to reaction mixtures). Such experiments would provide much clearer evidence of how the dihydroxylation proceeds. The current evidence does not lead directly to the conclusion that this FMO is inserting both oxygens via a hydroxylation mechanism.

Second, a clear list of the reducing reagents tested in the study needs to be included in the manuscript. The fact that only dithionite was capable of reducing the FAD and leading to hydroxylated product flies in the face of other catalyzed flavin-based hydroxylation reactions. Other FMOs frequently use NADPH or NADH; is there a plausible explanation of why NADPH cannot reduce QhpG. Is it possible to see a C4a hydroperoxyl-FAD in QhpG by itself or in a complex via UV at 382 nm? How fast will ~300 μM dithionite be consumed by O₂ at pH 8?

Third, QhpG forms a complex with QhpD and QhpC and does not interact well with crosslinked QhpC alone in the binding assay; it would be much more convincing and informative if the authors were able to show the docking of QhpC to QhpG in the presence of QhpD. The extensive discussion regarding the interaction of QhpC with QpgG is, thus, speculative and, perhaps, better presented in the SI.

Response to Reviewers' Comments

We thank the reviewers for critical reading of our manuscript and providing insightful comments and suggestions that have allowed us to greatly improve the quality of our manuscript. We have carefully reviewed all comments and have revised the manuscript accordingly. In particular, we have:

- Re-examined several reagents for reduction of the FAD bound to QhpG and showed the results (spectral data) in Supplementary Fig. 3.
- Added a new docking model of the QhpCDG ternary complex in Supplementary Fig. 12.
- Transferred two figures (Figs. 6 and 7) that were rather speculative to Supplementary Figs. 10 and 11. In addition, a speculative part of the text in the docking model construction was deleted, as it had been described in Methods.
- Conducted a new experiment for the QhpG reaction in H₂¹⁸O-buffer and added the result in the revised Fig. 3b.
- Assayed the remaining dithionite concentration after addition of O₂-saturated buffer (Supplementary Fig. 4).
- Compared the QhpG crystal structure with those of *Streptomyces venezuelae* alkylhalidase SvCmlS (Supplementary Fig. 7c) and a flavoprotein monooxygenase from *Schizosaccharomyces pombe* (Supplementary Fig. 9) to show similarities of their FAD-binding sites and absence of the NADPH-binding site in QhpG, respectively.

The revised text is colored red and is also included below our response to each comment. We provide a point-by-point response to all comments raised by the reviewers in blue font below (the line numbers refer to the revised manuscript unless otherwise stated):

Toshihide Okajima (on behalf of all co-authors)

Reviewer #1

This is an outstanding paper. It employs a wide range of approaches and challenging experiments to answer a longstanding and important question regarding the mechanism of CTQ biosynthesis in this enzyme. These results are relevant to the broad topic of unusual mechanisms of post-translational modifications to form protein-derived cofactors. It also describes a novel catalytic role for a flavoprotein. The evolutionary implications regarding the mechanisms tryptophylquinone biosynthesis in different enzymes and different organisms is also interesting and well stated.

I have a few comments.

- 1. Can the authors speculate on what might be the natural electron donor for the reaction, since it is not NADPH and dithionite was used in the study?*

We thank the reviewer for this important comment on the natural electron donor that we have not discussed in the present paper. We re-examined NADPH, NADH, and FADH₂ for reduction of the FAD bound to QhpG, but again none of them worked as reducing agents. Other natural small molecules, such as dihydrolipoate and reduced glutathione with lower reduction potentials than FAD, were also ineffective. Therefore, in the present study we used sodium dithionite that has been employed frequently in in-vitro experiments as an artificial reducing agent. To respond to the reviewer's comment, we added the following text in the revised manuscript (Main text, Discussion, lines 458–466):

“The physiological electron donor for the QhpG reaction is unknown at present. An electron-transfer protein existing in bacterial cells (e.g., ferredoxin, flavodoxin, and thioredoxin) may directly supply electrons for QhpG. Another possibility is that electrons supplied by an electron-transfer protein may be transferred to QhpG via QhpD that contains [4Fe-4S] clusters (one RS and two auxiliary clusters)^{13,14} within the QhpCDG ternary complex. These possibilities remain to be examined in future studies.”

2. *The reaction that was achieved is referred to as a single turnover. Does that mean that one molecule of enzyme could only form CTQ on one molecule of substrate? Was it possible to observe multiple turnovers with excess protein-complex substrate? If not why?*

We thank the reviewer for this insightful comment. As described in the paper, both of the QhpD-catalyzed Cys-Asp/Glu thioether bond formation and the QhpG-catalyzed Trp-hydroxylation (please note that the QhpG reaction is not CTQ formation) in QhpC proceed efficiently only in the QhpCDG ternary complex that is formed in the presence of stoichiometric amounts of these proteins and remains as a tight complex throughout the reactions (protein denaturation is always necessary for product identification). Preparation of an excess QhpD/crosslinked QhpC binary complex beyond QhpG is practically difficult because of the difficulty of preparing crosslinked QhpC without QhpG and so multiple turnovers of QhpG reaction cannot be tested experimentally. Therefore, we concluded that QhpG acts in a single-turnover manner, which is similar to the reactions of QhpD (ref. 14) and QhpE (ref. 15). In regard to the reviewer's comment, we added the following text in the revised manuscript (Main text, Discussion, lines 570–576):

“The single-turnover feature of the reactions of QhpD¹⁴, QhpE¹⁵, and QhpG (this paper) is consistent with the fact that the genes for these proteins are encoded within the same operon (*qhp*) as their substrate (QhpC) and they are expressed altogether under the control of the *n*-butylamine inducible transcriptional regulator¹¹. Thus, a single-use of each modifying enzyme is allowed in processing a single molecule of the substrate polypeptide, as reported for the ribosomally-synthesized and post-translationally-modified peptides (RiPPs) with various biological activities⁵³.”

3. *The evidence of CTQ formation was primarily based on mass spec analysis. It is convincing. Did the product with CTQ exhibit an absorbance spectrum with the characteristic CTQ spectral features? Was it possible to combine the gamma subunit product with the alpha and beta subunits and demonstrate catalytic activity?*

We thank the reviewer for this interesting comment. Although we did not measure an absorption spectrum of the QhpG-reaction product (crosslinked QhpC containing dihydroxylated-Trp (not CTQ)), the quinone-staining clearly indicated the absence of the quinone group in the reaction product (Supplementary Fig. 6). The dihydroxylated-Trp without an extended conjugate system is assumed to have no absorption band in a visible wavelength region. As reported previously (ref. 11), the gamma subunit precursor without CTQ may be translocated into periplasm, where the CTQ cofactor is finally formed probably by the aid of the alpha subunit that contains two heme *c* groups. It would be interesting to examine whether CTQ is generated in vitro by association of the gamma subunit precursor with the isolated alpha subunit and exhibits catalytic activity upon subsequent combination with the beta subunit. However, we have so far not succeeded separate expression of the alpha and beta subunits, and so the final generation of CTQ in the mature gamma subunit awaits further studies. To avoid confusion of the product of the QhpG reaction, we added the following sentence in the revised manuscript (Main text, Discussion, lines 556–560):

“By analogy with TTQ biogenesis in MADH, the final oxidation to CTQ of 6,7-dihydroxy-Trp in the crosslinked QhpC (pre- γ -subunit) produced by the QhpG reaction may be catalyzed by the two *c*-type hemes that are contained within the α -subunit, presumably after the periplasmic translocation of both of the pre- γ - and α -subunits¹¹.”

4. *The authors refer to the structure of GoxB in ref. 35 and correctly point out some distinctions. A somewhat remarkable point, which I do not think that they made, is that the GoxB structure was a homology model that used the structure of the Alkylhalidase CmlS. It received the best QMEAN Z score from a screen of the PDB database. This is the same protein (ref 18) that showed the highest structural similarity to QhpG. I find it amazing that both GoxB and QhpG are most similar to this halidase.*

We thank the reviewer for this insightful comment. We also already noticed that QhpG (this paper, crystal structure) and GoxB (ref. 42,

modeled structure) both show the highest structural similarity to alkylhalidase CmlS (ref. 18). This is why we included CmlS and GoxB/LodB as QhpG homologs in the multiple sequence alignment shown in Supplementary Fig. 6 (sequence similarities between CmlS and QhpG, 19%; CmlS and GoxB/LodB, ~18%; and QhpG and GoxB/LodB, 18%). To emphasize the remarkable structural similarity between QhpG and GoxB, we inserted the following text in the revised manuscript (Main text, Discussion, lines 512–514):

“...amine oxidation has been suggested⁴⁹. Both QhpG (crystal) and GoxB (model)⁴² show the highest structural similarity to alkylhalidase CmlS²⁰ with FAD bound to a nearly equivalent position and in an almost identical conformation (Supplementary Fig. 7c). However, FAD is bound loosely in GoxB⁴², but very tightly in QhpG,.....”

5. *Another point that I think is very interesting and should be given more emphasis is that QhpG inserts both oxygens into the Trp of CTQ. I believe that in all other known cases, TTQ in methylamine dehydrogenase, and CTQ in LodA and GoxA, the substrate for the modifying enzyme has the first hydroxyl present and only the second is added along with the formation of the crosslink.*

We thank the reviewer for this important comment. To emphasize this point, we inserted the following text in the revised manuscript (Main text, Discussion, lines 476–503):

“...is an unmodified Trp. Thus, QhpG inserts both oxygens into the Trp of CTQ, whereas in the biogenesis of TTQ in MADH and CTQ in LodA and GoxA, the substrate for the modifying enzyme has the first hydroxyl present and only the second is added along with the formation of the Trp–Trp or Trp–Cys crosslink. Formation of the initial....”

Reviewer #2

This manuscript by the Okajima group reports functional and structural analysis of a rather special flavoprotein (NADP) monooxygenase responsible for biosynthesis of cysteine tryptophylquinone (CTQ). Intriguingly, CTQ is generated from a protein-derived quinone cofactor.

The work reported here is comprehensive. It convincingly demonstrates the enzymatic activity of the flavoprotein. The high-resolution structure of QHNDH is well determined and informative. Functional analysis is sound and supports the main conclusion.

There are a few relatively minor concerns:

Abstract: “in the protein substrate containing triple intra-peptidyl sulfur-to-methylene carbon thioether bonds formed by a radical S-adenosylmethionine enzyme.”

Awkward phrasing

We thank the reviewer for this helpful comment. We revised the text as follows (Abstract, lines 22–24):

“...in the protein substrate containing triple intra-peptidyl crosslinks that were pre-formed by a radical S-adenosylmethionine enzyme within the ternary complex of these proteins.”

Line 207-208: “In view of the less extensive interactions between the two monomers, the dimerization appeared to be a crystallographic artifact.”

This conclusion should be more vigorously tested using such as Pissa analysis and even enzymatic experiment (Hills coefficient).

We thank the reviewer for this insightful comment. We performed PISA analysis (new ref. 17) and confirmed that there are no strong interactions enough to form a stable dimer in the protein interface. Unfortunately, however, an enzymatic experiment for determination of the Hill coefficient is impracticable as the QhpG reaction proceeds only in the QhpCDG ternary complex. Molecular weight determination by a gel-filtration method also suggested that QhpG is a monomer protein in solution. To support this conclusion, we inserted the following text in the revised manuscript (Main text, Results, Crystal structure of QhpG, lines 264–268):

“However, PISA analysis¹⁷ indicated that there are no strong interactions enough to form a stable dimer in the protein interface. Additionally, molecular weight determination by a gel-filtration method suggested that QhpG is a monomer protein in solution, as described above. Thus, the dimerization is assumed to be a crystallographic artifact.”

Ramachandran Outlier 0.2% - This is not insignificant for protein of this size and structure of this resolution. Authors should examine these outliers and see whether they have any potential connection to amino acid modifications.

We thank the reviewer for this crystallographic comment. The outlier residues in Ramachandran plot are only Val119 in chains A and B (2 aa/858 aa). Val119 is contained in a short turn (Asp118–Gly120), in which the main chain carbonyl group of Asp118 is hydrogen-bonded to the main chain amide group of Gly120, stabilizing the turn structure with relatively high thermal factors. We also confirmed that Val119 has no potential connection to amino acid modifications. To explain the Ramachandran outlier, we added the following description as a footnote (*b*) in Supplementary Table 2:

“The outlier residues are only Val119 in chains A and B (2/858). Val119 is contained in a short turn (Asp118–Gly120), in which the main chain carbonyl group of Asp118 is hydrogen-bonded to the main chain amide group of Gly120, stabilizing the turn structure with relatively high thermal factors.

Lines 85-86: “three intra-peptidyl thioether bonds formed between Cys and Asp or Glu residues”

I'm not sure I understand this. Formation of a thioether from Cys and Glu/Asp (See Figure 1B) involves the loss of 2 x H, or 2 Da, so three thioether bonds would involve a loss of 6 Da, not 28.

We thank the reviewer for this comment on the mass value estimation and are very sorry for the ambiguous description. Compared to the matured γ -

subunit containing CTQ and three Cys-to-Asp/Glu thioether bonds, the QhpC polypeptide produced in the $\Delta qhpG$ mutant strain does not contain CTQ (without 2 oxygen atoms (in an *ortho* quinone form) and a Cys–Trp crosslink; $16 \times 2 - 2 - 2 = 28$). This result is very important to conclude that the QhpG enzyme participates in the incorporation of 2 oxygen atoms into the QhpC polypeptide. To clarify this point, we inserted the following phrase (in parentheses) (Main text, Results, Analysis of quinone-less γ -subunit in inactive QHNDH, lines 109–110):

“...unmodified Cys and Trp residues (without 2 oxygen atoms and a Cys–Trp crosslink contained in CTQ of γ -subunit) (Supplementary Fig. 1a).”

Line 168, “*m/z*, 2054.673 \pm 0.213”

Significant figures should be to one decimal place.

We thank the reviewer for this precise comment. We corrected all *m/z* values to one decimal place in the revised manuscript (Main text, Results, Determination of catalytic activity of QhpG, lines 203, 204, 207, and 208).

Lines 287-288: “*regiospecifically insert two hydroxyl groups into a single substrate,*”

Could have better context. There are a few more examples of enzymes that perform multiple, regio- and stereospecific oxidation reactions.

Some P450 examples:

*Sky26, a monooxygenase that performs 3 regiospecific hydroxylations:
Pohle, S., Appelt, C., Roux, M., Fiedler, H.-P., and Süßmuth, R. D. (2011)
Biosynthetic Gene Cluster of the Non-ribosomally Synthesized
Cyclodepsipeptide Skyllamycin: Deciphering Unprecedented Ways of
Unusual Hydroxylation Reactions. 133, 6194–6205*

DoxA, doxorubicin biosynthesis (2x hydroxylation)

MycG, mycinamycin biosynthesis (1 hydroxylation, 1 epoxidation)

TamI, tirandamycin B biosynthesis (2 x hydroxylation, 1 epoxidation).

(above 3 examples cited in review by Greule, A., Stok, J. E., De Voss, J. J., and Cryle, M. J. (2018) Unrivalled diversity: the many roles and reactions of bacterial cytochromes P450 in secondary metabolism. Nat. Prod. Rep. 35, 757–791)

Also related is the flavin dependent brominase Bmp5, which performs two successive regiospecific bromination reactions:

1. Agarwal, V., Gamal, El, A. A., Yamanaka, K., Poth, D., Kersten, R. D., Schorn, M., Allen, E. E., and Moore, B. S. (2014) Biosynthesis of polybrominated aromatic organic compounds by marine bacteria. Nat Chem Biol. 10, 640–647

We thank the reviewer for this valuable comment. We incorporated all these additional examples into the revised manuscript with slight revision of the text as follows (Main text, Discussion, lines 377–398):

“Such enzymes that regio- and stereo-specifically insert two or more hydroxyl groups into a single substrate have been reported for several cytochrome P450 monooxygenases²⁹; 5-epiaristolochene 1,3-dihydroxylase (EAH) involved in capsidiol biosynthesis in plant³⁰ (2× hydroxylation), DoxA²⁹ involved in doxorubicin biosynthesis (2× hydroxylation), Sky32³¹ involved in biosynthesis of a cyclic depsipeptide skyllamycin A (3× hydroxylation), TamI²⁹ involved in tirandamycin B biosynthesis (2× hydroxylation, 1 epoxidation), and MycG²⁹ involved in mycinamycin biosynthesis (1 hydroxylation, 1 epoxidation). A microbial nonheme Fe^{II} α-ketoglutarate-dependent oxygenase (named OrfP) involved in antibiotic (streptothricin-F) biosynthesis³² also inserts two hydroxyl groups into a single substrate. However, within the flavoprotein monooxygenase (FMO) family, there is no precedent for the enzyme that catalyzes dihydroxylation, although there is a related enzyme brominase (Bmp5)³³, which performs two successive regiospecific bromination reactions.

Referring to the mechanisms of EAH and OrfP, both....”

Lines 299-301: “We speculate that the indole ring C7 position is the first hydroxylation site because of elevated electronegativity by the formation of a hydrogen bond between the indole ring N1 and FAD O4 atoms.”

Replace “...elevated electronegativity.....” with “elevated nucleophilicity of this position induced....”

We thank the reviewer for this kind suggestion. We revised the text as suggested (Main text, Discussion, lines 407–412):

“...because of elevated nucleophilicity of this position induced by the formation....”

Lines 365-368: “The protein machinery consisting of the two enzymes swallows a raw material (QhpC) and crafts a decorated product with a complicated structure (pre- γ -subunit), like magician’s hands manipulating a linear rope and changing it into a multi-knotted lariat with a pendant.”

Purple prose. I would delete this. The authors should note that this is very analogous to the biosynthesis of RiPPs.

We thank the reviewer for this valuable comment. We deleted this sentence and replaced with the following text describing the similarity of the CTQ biogenesis performed by the enzymes encoded in the *qhp* operon to those of the ribosomally synthesized and post-translationally modified peptides (RiPPs) (Main text, Discussion, lines 554–560):

“The single-turnover feature of the reactions of QhpD¹⁴, QhpE¹⁵, and QhpG (this paper) is consistent with the fact that the genes for these proteins are encoded within the same operon (*qhp*) as their substrate (QhpC) and they are expressed altogether under the control of the *n*-butylamine inducible transcriptional regulator¹¹. Thus, a single-use of each modifying enzyme is allowed in processing a single molecule of the substrate polypeptide, as

reported for the ribosomally-synthesized and post-translationally-modified peptides (RiPPs) with various biological activities⁵³.”

Reviewer #3

This manuscript reports the first structural and functional characterization of QhpG as a catalyst for the biogenesis of CTQ within QhpC, one of three subunits that constitute the mature, multi-subunit enzyme quinohemoprotein amine dehydrogenase(QHNDH). In the past, the hydroxylation of Trp in QhpC was proposed to occur via a flavin-dependent monooxygenase (FMO) based on sequence similarity. Using a combination of biochemical, biophysical and structural biology methods, the authors in this manuscript discover that the qhpG gene encoded protein that uses one equivalent of FAD to conduct dihydroxylation on the Trp residue within QhpC, in the presence of a reducing reagent and subsequent to the formation of interchain thioether linkages catalyzed by QhpD. The interaction between substrate QhpC and QhpG before and after modification by QhpD suggests that the QhpG modifies the crosslinked QhpC. The authors have obtained a fairly high-resolution X-ray crystal structure of QhpG with FAD bound in the active site, and their computational docking suggests that the active site can accommodate the substrate QhpC possibly via electrostatic interaction.

Although FMO has been reported in numerous biosynthetic pathways, this is the first study that suggests that FMO modifies the Trp in the maturation of CTQ cofactor in a dihydroxylation manner. If the mechanism as proposed is correct, it is rare for FMO. However, other than the demonstration of a dihydroxy-Trp product, there are relatively few data that can inform the mechanism presented by the authors. Three key issues need to be addressed, before the paper can be recommended for publication.

First, the paper would be considerably strengthened by experiments that can distinguish whether the two oxygens in the modified Trp come from O₂ vs H₂O (via the addition of ¹⁸O₂-saturated buffer or H₂¹⁸O to reaction mixtures). Such experiments would provide much clearer evidence of how

the dihydroxylation proceeds. The current evidence does not lead directly to the conclusion that this FMO is inserting both oxygens via a hydroxylation mechanism.

We thank the reviewer for this extremely important comment. We carried out the QhpG reaction by first anaerobically reducing FAD (in the QhpCDG complex) with dithionite in the buffer replaced with H₂¹⁸O (≥ 98 atom% ¹⁸O) then adding normal ¹⁶O₂-saturated H₂¹⁸O-buffer for oxygenation. We found that the reaction product contains two ¹⁶O atoms but no ¹⁸O atom, supporting that the oxygen atoms inserted into the CTQ-precursor Trp are not derived from solvent H₂O. We incorporated this new result into the revised manuscript along with the mass spectroscopic data as follows (Main text, Results, Determination of catalytic activity of QhpG, lines 211–215; and Fig. 3). A few sentences for the procedure of the reaction in H₂¹⁸O-buffer were also added in the Methods section (Main text, Methods, lines 624–629):

(lines 211–215)

“Furthermore, when the initial anaerobic reduction with dithionite was done in the H₂¹⁸O-buffer and then the QhpG reaction was started by the addition of ¹⁶O₂-saturated H₂¹⁸O-buffer, the reaction product contained two ¹⁶O atoms but no ¹⁸O atom (Fig. 3b), supporting that the oxygen atoms inserted into the CTQ-precursor Trp are not derived from solvent H₂O.”

(lines 624–629)

“For the reaction in H₂¹⁸O, buffer C dissolving QhpG and the QhpCD binary complex was exchanged with H₂¹⁸O (≥ 98atom% ¹⁸O; Taiyo Nissan Co., Tokyo, Japan) by repeating concentration and dilution three times using an Amicon 10K concentrator. The ¹⁶O₂-saturated buffer C was also prepared from the H₂¹⁸O-buffer C. The QhpG reaction and MS analysis of the product were conducted as described above.”

We also conducted the QhpG reaction by directly introducing ¹⁸O₂ gas to the dithionite-reduced reaction mixture in an anaerobic chamber filled with 99.999% N₂ gas, since the commercially available ¹⁸O₂ gas (0.5 liter, enclosed in a 0.35-liter closed container; only 0.15 liter is usable at

atmospheric pressure) was insufficient for preparing $^{18}\text{O}_2$ -saturated buffer and filling inside the reaction tube. In this case, however, no reaction product (showing expected $\mathbf{d}+36$ peak) was detected. We assume that the introduced $^{18}\text{O}_2$ was consumed by dithionite or immediately diffused into the anaerobic chamber before reacting with FADH_2 of QhpG in the QhpCDG ternary complex. Presumably, more rigorous control of the reaction conditions is needed for repeating the $^{18}\text{O}_2$ experiment, if necessary.

Second, a clear list of the reducing reagents tested in the study needs to be included in the manuscript. The fact that only dithionite was capable of reducing the FAD and leading to hydroxylated product flies in the face of other catalyzed flavin-based hydroxylation reactions.

We thank the reviewer for this important comment. The reducing reagents tested for anaerobic reduction of the QhpG-bound FAD in this study were NADPH, NADH, FADH_2 , and sodium dithionite (they all were listed in the revised manuscript). We re-examined NADPH, NADH, and FADH_2 (each 0.1 mM) for reduction of the QhpG-bound FAD, but again none of them worked as reducing reagents (free FADH_2 neither reduced nor replaced the bound FAD). We incorporated this result as a figure in the revised Supplementary Information (Supplementary Fig. 3). Two other small biomolecules, dihydrolipoate and reduced glutathione with lower reduction potentials than FAD, were also tested but found to be ineffective (Supplementary Fig. 3). Incorporating these data, we revised the manuscript as follows (Main text, Results, Determination of catalytic activity of QhpG, lines 186–193):

“...., the purified QhpG was first anaerobically incubated with several reducing reagents: NADPH, NADH, FADH_2 , and sodium dithionite. Two other small biomolecules, dihydrolipoate and reduced glutathione with lower reduction potentials than FAD, were also tested. Beside the artificial reductant (sodium dithionite) (Supplementary Fig. 2b), none of the physiological reagents reduced the QhpG-bound FAD without affecting its absorption spectrum (Supplementary Fig. 3b–f). Free FADH_2 neither reduced nor replaced the bound FAD. Therefore, after reducing....”

Other FMOs frequently use NADPH or NADH; is there a plausible explanation of why NADPH cannot reduce QhpG.

The inability of NAD(P)H for reduction of the QhpG FAD may be explained by the absence of an NAD(P)H-binding site in QhpG, unlike other FMOs, which often utilize NAD(P)H as an electron donor. To help understanding the structural difference between QhpG and other typical NAD(P)H-dependent FMOs, we compared the crystal structures of QhpG (this study) and FMO from *Schizosaccharomyces pombe* (SpFMO, PDB ID: 2GV8) (new ref. 24) in Supplementary Fig. 9 of the revised Supplementary Information. It is evident that the NADPH-binding region of SpFMO and the corresponding region of QhpG have considerably different secondary structural motifs (a parallel five-stranded β -sheet, an antiparallel three-stranded β -sheet, and four helices vs. an antiparallel five-stranded β -sheet and two helices). To describe a structural difference of QhpG from other NAD(P)H-dependent FMOs, we added the following text in the revised manuscript (Main text, Results, Crystal structure of QhpG, lines 282–290):

“In marked contrast to the well conserved FAD-binding site, QhpG has considerably different secondary structural motifs in the region corresponding to the NAD(P)H-binding region of other flavin monooxygenases (FMOs). For example, an FMO from *Schizosaccharomyces pombe* (PDB ID: 2GV8)²⁴ has the NADPH-binding region consisting of a parallel five-stranded β -sheet, an antiparallel three-stranded β -sheet, and four α -helices, with a consensus sequence for the nucleotide-binding loop (GXSXXA), whereas the corresponding region of QhpG consists of an antiparallel five-stranded β -sheet and two α -helices without the consensus sequence (Supplementary Fig. 9).”

Is it possible to see a C4a hydroperoxyl-FAD in QhpG by itself or in a complex via UV at 382 nm?

We performed a spectrophotometric study to examine the formation of the C4a-hydroperoxyl-FAD intermediate by addition of O₂-saturated buffer to the dithionite-reduced FAD bound to QhpG. The absorption bands in the visible region (350–500 nm), which are characteristic of the oxidized form

of FAD, appeared rapidly (within 3 seconds) upon addition of twice volumes of O₂-saturated buffer without forming an intermediary absorption peak at 382 nm, as shown in the revised Supplementary Fig. 3a. This suggests that the C4a-hydroperoxyl-FAD intermediate, if it were formed in QhpG, has a very short lifetime under the conditions (even in the absence of the protein substrate (crosslinked QhpC)). Regarding this result, we inserted a short sentence in the revised manuscript (Main text, Results, Purification and characterization of QhpG, lines 132–133):

“Addition of O₂-saturated buffer to the dithionite-reduced QhpG resulted in rapid re-oxidation of the bound FAD (Supplementary Fig. 3a).”

How fast will ~300uM dithionite to be consumed by O₂ at pH 8?

We are very sorry for incorrect description of the initial concentration of sodium dithionite (1 mM) in the QhpG reaction (before addition of twice volumes of O₂-saturated buffer (Methods, Assay of QhpG activity, lines 519–523 of the original manuscript). The correct concentration of dithionite in the first anaerobic incubation was 3 mM in a 0.25-ml reaction mixture. So, the reviewer’s comment should be read ‘How fast will ~1 mM dithionite be consumed by O₂ at pH 8?’. We assayed the remaining dithionite concentration in the reaction mixture after addition of O₂-saturated buffer (in the absence of the proteins (QhpCDG ternary complex); *n* = 4). As shown in the revised Supplementary Fig. 4, immediately after addition of twice volumes of O₂-saturated buffer (0.5 ml) to the initial dithionite solution (3 mM, 0.25 ml), the dithionite concentration dropped to about 0.65 mM and then decreased exponentially to about 0.1 mM after 60 min. Accordingly, we revised the text as follows (Main text, Results, Determination of catalytic activity of QhpG, lines 195–197):

“...under an atmospheric condition until the initially added dithionite was mostly consumed by air (Supplementary Fig. 4). The reaction product...”

In the Methods section, we corrected the dithionite concentration (Main text, Methods, Assay of QhpG activity, lines 733–734):

“...was incubated with 3 mM sodium dithionite...”

Third, QhpG forms a complex with QhpD and QhpC and does not interact well with crosslinked QhpC alone in the binding assay; it would be much more convincing and informative if the authors were able to show the docking of QhpC to QhpG in the presence of QhpD. The extensive discussion regarding the interaction of QhpC with QpgG is, thus, speculative and, perhaps, better presented in the SI.

We thank the reviewer for this comment, but we wonder why the reviewer interpreted that QhpG does not interact well with crosslinked QhpC alone in the binding assay. Although the QhpG reaction does not proceed with the crosslinked QhpC alone (Fig. 3a, *bottom* panel), both BLI assay and mobility shift assay demonstrate that QhpG does interact well with crosslinked QhpC (Fig. 2, panel *c* and *d*) with a dissociation constant (28 nM) much smaller than the linear QhpC (Supplementary Table 1). Moreover, crystal structures of both QhpG (this paper) and γ -subunit (mature form of QhpC) from the same bacterium (*Pseudomonas putida*) (ref. 10) are available for more reliable docking model building than using a modeled structure, which prompted us to construct first the docking model for the QhpCG binary complex. However, we agree that the extensive discussion regarding the interaction of QhpC with QhpG may be speculative. So, we decided to transfer two relevant figures (Figs. 6 and 7 in the original manuscript) to Supplementary Information (Supplementary Figs. 10 and 11) along with deleting a part of the text (Main text, Results, Construction of crosslinked QhpC-QhpG docking model, lines 263–267 of the original manuscript), which is mostly described in the Methods section (Main text, Methods, Construction of docking models, lines 819–829). To emphasize the rationality of constructing the docking model of crosslinked QhpC-QhpG (binary complex), we revised the beginning of this section as follows (Main text, Results, Construction of docking models of binary and ternary complexes, lines 325-326):

“Based on the biochemical evidence showing the significant interaction between QhpG and crosslinked QhpC described above, we...”

As for construction of the structure model of the QhpCDG ternary complex, we initially hesitated it because of the absence of the crystal

structure of QhpD. However, we made an effort to construct it using a structure model of QhpD from *Ps. putida*. Thus, a QhpD model was first created by SWISS-MODEL, a homology-modelling server (<https://swissmodel.expasy.org>), using a sactonine bond-forming enzyme CteB from *Clostridium thermocellum* (PDB ID: 5WGG) as a template. Subsequently, we could obtain a possible model of the QhpCDG ternary complex with ZDOCK by docking the above QhpD model structure to the model of the crosslinked QhpC-QhpG binary complex (Supplementary Fig. 12). We added the following text in the revised manuscript (Main text, Results, Construction of docking models of binary and ternary complexes, lines 355–373):

“For construction of the docking model of the QhpCDG ternary complex, a structure model of QhpD had to be first generated using the SWISS-MODEL homology-modelling server (<https://swissmodel.expasy.org>)²⁷, because of the absence of its crystal structure. In the modeling, a radical SAM enzyme, a sactonine bond-forming enzyme CteB from *Clostridium thermocellum* (PDB ID: 5WGG)²⁸ was auto-selected as the template based on sequence homology. Subsequently, we could obtain a possible model of the QhpCDG ternary complex with ZDOCK by docking the above QhpD model structure to the model of the QhpCG binary complex (Supplementary Fig. 12). As reported previously for QhpD from *Pa. denitrificans*¹³, the structure model of *Ps. putida* enzyme also has a large groove with sufficient space to accommodate the core QhpC polypeptide containing several negatively charged residues. The structure model of the ternary complex shows that QhpD binds the crosslinked QhpC at the large groove and from the opposite side of QhpC involved in the interaction with QhpG. Altogether, the crosslinked QhpC may be sandwiched by QhpD and QhpG and serves as the common protein substrate for these two enzymes, to undergo efficient and successive crosslinking and Trp-dihydroxylation reactions.”

REVIEWERS' COMMENTS

Reviewer #1 (Remarks to the Author):

The authors have addressed my few concerns and revised the paper accordingly. In my opinion they have also done a good job of addressing the comments of the other reviewers. These revisions have improved what was already an excellent paper.

Reviewer #2 (Remarks to the Author):

The authors have made good efforts and addressed my minor concerns.

Reviewer #3 (Remarks to the Author):

The authors have carefully addressed all of my questions and concerns. Publication of the revised manuscript is recommended.